Corrected: Author correction

# Sensory innervation in porous endplates by Netrin-1 from osteoclasts mediates PGE2-induced spinal hypersensitivity in mice

Shuangfei Ni[1,2], Zemin Ling[1,3], Xiao Wang [1], Yong Cao[1,2], Tianding Wu[1,2], Ruoxian Deng[1], Janet L. Crane[1,4], Richard Skolasky[1], Shadpour Demehri[5], Gehua Zhen[1], Amit Jain[1], Panfeng Wu[1], Dayu Pan [1], Bo Hu[1], Xiao Lyu[1], Yusheng Li[1], Hao Chen[1], Huabin Qi[1], Yun Guan[6], Xinzhong Dong[7], Mei Wan[1], Xuenong Zou[3], Hongbin Lu[8], Jianzhong Hu[2]* & Xu Cao[1]*

Spinal pain is a major clinical problem, however, its origins and underlying mechanisms remain unclear. Here we report that in mice, osteoclasts induce sensory innervation in the porous endplates which contributes to spinal hypersensitivity in mice. Sensory innervation of the porous areas of sclerotic endplates in mice was confirmed. Lumbar spine instability (LSI), or aging, induces spinal hypersensitivity in mice. In these conditions, we show that there are elevated levels of PGE2 which activate sensory nerves, leading to sodium influx through Na$_v$ 1.8 channels. We show that knockout of PGE2 receptor 4 in sensory nerves significantly reduces spinal hypersensitivity. Inhibition of osteoclast formation by knockout *Rankl* in the osteocytes significantly inhibits LSI-induced porosity of endplates, sensory innervation, and spinal hypersensitivity. Knockout of *Netrin-1* in osteoclasts abrogates sensory innervation into porous endplates and spinal hypersensitivity. These findings suggest that osteoclast-initiated porosity of endplates and sensory innervation are potential therapeutic targets for spinal pain.

[1] Department of Orthopaedic Surgery, The Johns Hopkins University School of Medicine, Baltimore, MD 21205, USA. [2] Department of Spine Surgery, Xiangya Hospital, Central South University, Changsha 410008, P. R. China. [3] Guangdong Provincial Key Laboratory of Orthopedics and Traumatology, Department of Spinal Surgery, The First Affiliated Hospital of Sun Yat-sen University, Guangzhou 51008, P. R. China. [4] Department of Pediatrics, The Johns Hopkins University School of Medicine, Baltimore, MD 21205, USA. [5] Department of Radiology and Radiological Science, The Johns Hopkins University School of Medicine, Baltimore, MD 21205, USA. [6] Department of Anesthesiology and Critical Care Medicine, The Johns Hopkins University School of Medicine, Baltimore, MD 21205, USA. [7] Department of Neuroscience, Neurosurgery, and Dermatology, Center of Sensory Biology, The Johns Hopkins University School of Medicine, Howard Hughes Medical Institute, Baltimore, MD 21205, USA. [8] Department of Sports Medicine, Xiangya Hospital, Central South University, Changsha 410008, P. R. China. *email: jianzhonghu@hotmail.com; xcao11@jhmi.edu

Low back pain (LBP) is a common health problem, which most people (80%) experience at some point, especially in older adults[1–4]. In the USA alone, the direct and indirect costs associated with LBP surpass $90 billion per year, with similar adjusted rates in other countries[5]. In total, 90% of LBP is non-specific LBP, which has no apparent pathoanatomical cause[6,7]. Several lumbar structures, such as intervertebral disc, facet joints, are plausible sources of nonspecific LBP, but the pain can not be reliably attributed to those structures by clinical tests[8–10]. Importantly, intervertebral disc (IVD) degeneration is frequently observed in asymptomatic patients, indicating that disc degeneration, per se, is not painful in some patients[11,12]. Hence, identifying the source of LBP and related mechanisms is essential to develop effective treatments for LBP.

The positive association between vertebral endplate signal changes (i.e., Modic changes) and LBP has been shown by magnetic resonance imaging (MRI) examination[13,14]. Modic changes are common MRI findings in patients with nonspecific LBP. It is believed to be a factor independently associated with the increased risk of LBP[15–17]. The size of Modic change lesions positively correlates with LBP[18]. Histological analysis further showed that the presence of endplate lesions was associated with LBP[19]. During aging, endplates undergo ossification with elevated osteoclasts and become porous[20–23]. Histological and micro CT analysis further revealed that sclerotic endplates are highly porous[22]. Progressively more porous endplates with narrowed IVD space are characteristics of spinal degeneration[22,24]. Since pain is produced by nociceptors, LBP may be caused by sensory innervation into endplates. Moreover, nerve density was higher in porous endplates than in normal endplates or degenerative nucleus pulposus[25]. Zoledronic acid and denosumab, drugs that inhibit osteoclast activities, have shown analgesic effects in patients with Modic changes associated LBP[26,27], with the implication of a potential role of osteoclast activity in sensory nerve innervation.

Prostaglandin E2 (PGE2) is an inflammatory mediator released at the focal inflamed tissue, and a neuromodulator that alters neuronal excitability. Four types of G-protein-coupled EP receptors (EP1–EP4) mediate the functions of PGE2. EP4 receptor is considered the primary mediator of PGE2-evoked inflammatory pain hypersensitivity and sensitization of sensory neurons[28,29]. We recently reported that PGE2, produced from arachidonic acid by the enzymatic activity of cyclooxygenase 2 (COX-2), during bone remodeling activates PGE2 receptor 4 (EP4) in $CGRP^+$ sensory nerves to tune down sympathetic tones further inducing osteoblastic differentiation of MSCs[30]. Specific EP4 receptor antagonists could reduce acute and chronic pain, including osteoarthritis pain[28,31–33]. The tetrodotoxin-resistant (TTX-R) sodium channel $Na_v1.8$ is a potential drug target for pain. $Na_v1.8$ is expressed primarily in small and medium-sized dorsal root ganglion (DRG) neurons and their fibers[34–36]. PGE2 could modulate the TTX-R sodium current in DRG neurons and promote $Na_v1.8$ trafficking to the cell surface[37,38].

In this study, we sought to demonstrate that osteoclasts initiate porosity of endplates with sensory innervation into porous areas. Our data showed that Calcitonin Gene-Related Peptide positive ($CGRP^+$) nociceptive nerve fibers and blood vessels were increased in the cavities of sclerotic endplates. The elevated PGE2 in porous endplates induces sodium influx into the cells to stimulate sensory nerves that leads to spinal pain. Inhibition of osteoclast activity attenuated sensory innervation in porous endplates and pain behavior.

## Results

**Development of hyperalgesia in LSI and aged mouse models.** Lumbar spine instability (LSI) was established in mice as a spine

degeneration model for spine pain behavior testing[20,39,40]. We first measured the vocalization threshold in response to force applied on the L4/L5 disc region. Pressure tolerance decreased significantly at 4, 8, and 12 weeks after LSI surgery relative to mice that underwent sham surgery (Fig. 1a), indicating the development of low back pressure hyperalgesia. Then, we monitored spontaneous activity to indicate the potential effect of spinal pain, including distance traveled, maximum speed of movement, mean speed of movement, and active time per 24 h, though they are not specific for spinal pain behaviors. The results revealed that each measure of spontaneous activity decreased significantly at 4 and 8 weeks after LSI surgery relative to sham surgery (Fig. 1b–e). Moreover, we assessed mechanical hyperalgesia of the hind paw as referred pain by performing von Frey analysis, as a secondary indicator of symptomatic LBP. We found that the paw withdraw frequency increased significantly from 2 to 12 weeks after LSI surgery (Fig. 1f, g). However, we observed no response to the straight leg-raising test when recording the number of vocalizations during five leg stretch-and-lifts in either LSI or sham surgery mice. This indicates that nerve root compression is not involved in the hyperalgesia developed after LSI surgery. The results of these pain behavior tests suggest that spine instability induces the development of hyperalgesia.

In parallel, we evaluated symptomatic LBP during aging in these behavior tests. Similarly, the threshold of pressure tolerance (Fig. 1h) and spontaneous activity (Fig. 1i–k) decreased significantly in aged mice (age 20 months) relative to young mice (age 3 months). The mechanical hyperalgesia of the hind paw increased significantly in aged mice relative to young mice (Fig. 1l, m). Together, these data indicate that, as in the LSI mouse model, aging also induces spine hyperalgesia.

**Sensory innervation in endplates in LSI and aged mouse models.** We showed previously an increase in osteoclasts at the onset of endplate sclerosis[20]. Therefore, we evaluated the potential role of osteoclasts in the sensory innervation of endplates. Tartrate-resistant acid phosphatase (TRAP) staining demonstrated that the number of $TRAP^+$ osteoclasts in endplates increased significantly at 2 weeks after LSI, and remained at a high level until 8 weeks after LSI surgery (Fig. 2a, b). Large bone marrow cavities were generated in sclerotic endplates by osteoclastic bone resorption in mice after LSI surgery, whereas, the cartilaginous endplates were maintained in sham surgery mice (Fig. 2a). Immunofluorescent staining revealed that the significant increase of CGRP, the marker of peptidergic nociceptive C nerve fibers in the porous endplates, began at 2 weeks and continued to increase until 8 weeks after LSI surgery (Fig. 2c, d), but there were no detectable $CGRP^+$ nerves in the endplates of sham surgery mice (Fig. 2c, d). Interestingly, the nociceptive nerve fibers were localized primarily adjacent to the bone surface (Fig. 2c). The co-staining of PGP9.5, the broad marker of nerve fibers with CGRP, further validated the nociceptive innervation of the endplates after LSI surgery (Fig. 2e). However, the nonpeptidergic subtype of $IB4^+$ C nerve fibers was not detected in either LSI or sham surgery mice (Fig. 2f), suggesting $CGRP^+$ nerve fibers as the primary nociceptive C nerve fibers in the endplates. Importantly, $CD31^+EMCN^+$ blood vessels were also growing into the porous endplates after LSI surgery, along with the sensory innervation of the endplates (Supplementary Fig. 1A, B).

To determine whether spine degeneration during aging could induce sclerosis and sensory innervation in vertebral endplates, we analyzed the caudal endplates of L4/5 from aged mice and young mice. The porosity of endplates in aged mice increased significantly relative to young mice, as determined by

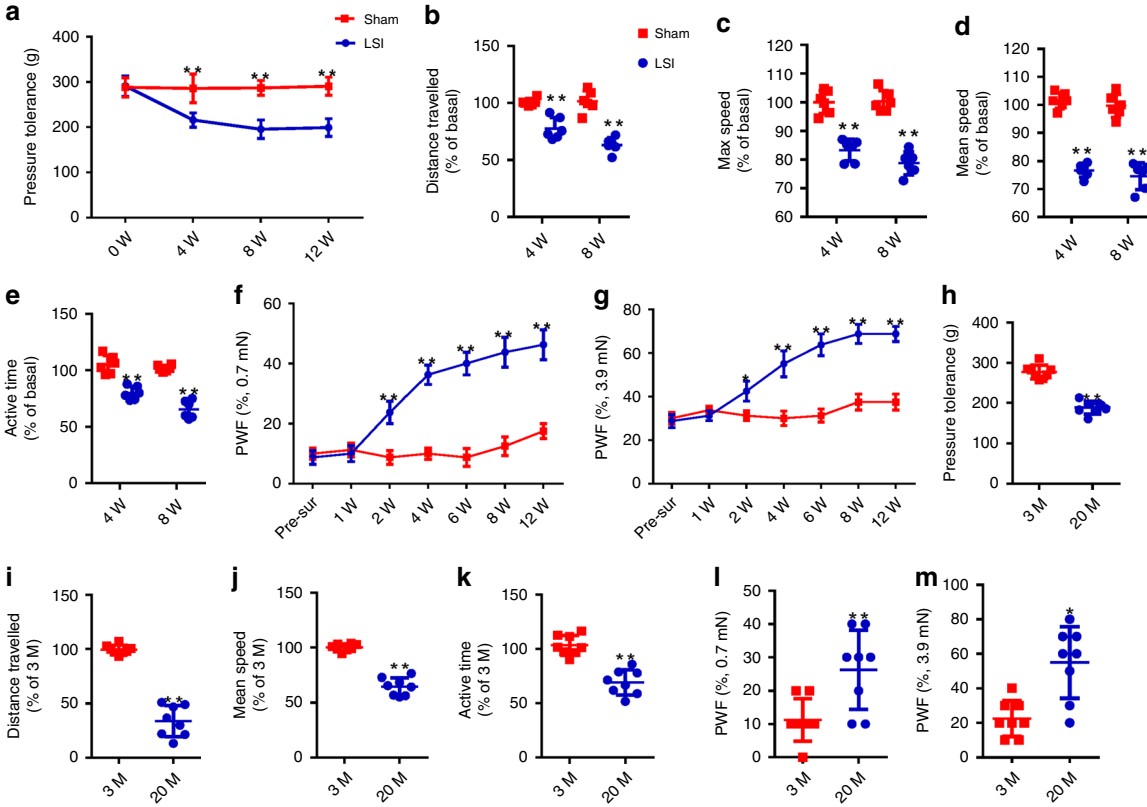

**Fig. 1 Symptomatic spinal pain behavior in the LSI model and aged mice. a** Pressure hyperalgesia of the lumbar spine was assessed as the force threshold to induce the vocalization by a force gauge after LSI or sham surgery. **b–e** Spontaneous activity analysis, including distance traveled (**b**), maximum speed (**c**), mean speed (**d**), and active time (**e**) on the wheel per 24 h, determined by the percentage of sham surgery mice at the corresponding time points. **f, g** The hind paw withdrawal frequency responding to mechanical stimulation (von Frey, 0.7 mN and 3.9 mN) after LSI or sham surgery. *$p < 0.05$, **$p < 0.01$ compared with the sham surgery mice at the corresponding time points. $n = 6$ per group (**a–g**). **h** Pressure hyperalgesia of the low back in 20-month-old or 3-month-old mice. **i** The distance traveled, (**j**) mean speed, and (**k**) active time on the wheel per 24 h in 20-month-old mice determined by the percentage of 3-month-old mice. **l, m** The hind paw withdrawal frequency responding to mechanical stimulation (von Frey, 0.7 mN and 3.9 mN) in 20-month-old or 3-month-old mice. *$p < 0.05$, **$p < 0.01$ compared with the 3-month-old mice. $n = 8$ per group (**h–m**). Statistical significance was determined by multifactorial ANOVA, and all data are shown as means ± standard deviations. PWF paw withdraw frequency. Source data are provided as a Source Data file.

three-dimensional microcomputed tomography (µCT) analysis (Fig. 3a–c). The µCT analysis of vertebral trabecular bone demonstrated that the trabecular bone volume/total volume (BV/TV) and trabecular bone number (Tb.N) of L5 vertebrae decreased significantly in 20-month-old mice relative to 3-month-old mice, while the trabecular bone thickness (Tb.Th) and trabecular bone separation distribution (Tb.Sp) did not change significantly (Supplementary Fig. 2A–E). Safranin O and fast green staining demonstrated that the green-stained bone matrix surrounded the cavities in endplates of aged mice (Fig. 3d, top and middle), suggesting endochondral ossification. Endplate scores, which are a histologic assessment of pathological changes, such as bony sclerosis, structure disorganization, and neovascularization, were significantly higher in aged mice than that in young mice (Fig. 3e). Interestingly, high levels of TRAP+ osteoclasts were observed in the endplates of aged mice, whereas TRAP+ osteoclasts were rarely detected in the endplates of young mice (Fig. 3c, bottom and 3f). Immunostaining of CGRP showed increased aberrant innervation of peptidergic nociceptive C nerve fibers in the porous endplates of aged mice (Fig. 3g, h). The co-staining of PGP9.5 and CGRP further confirmed that the endplates were innervated by nociceptive nerve fibers (Fig. 3i). Similar to our findings in LSI mice, CD31+EMCN+ blood vessels were detected in the endplates, along with CGRP+ nerve fibers

during aging (Supplementary Fig. 3A, B), indicating active ossification of the endplates.

To examine the potential involvement of sclerosis and sensory innervation of the endplates with pain behavior, we evaluated the pathological changes in the endplates of the lower lumbar spines from patients with or without LBP history. Severe endplate lesions were observed in patients with a history of frequent LBP, whereas the cartilaginous structure was preserved in patients without a history of frequent LBP, despite disc herniation (Supplementary Fig. 4A). The increased endplate scores were also observed in patients with a history of frequent LBP (Supplementary Fig. 4B). However, the patients with the history of frequent LBP are older than the ones without the history of frequent LBP (Supplementary Table 1). TRAP staining showed that abundant TRAP+ osteoclasts localized at the bone surface in the sclerotic endplates (Supplementary Fig. 4C). Immunofluorescence staining revealed that CGRP+PGP9.5+ nociceptive nerve fibers grown into the porous areas of sclerotic endplates of patients with LBP history (Supplementary Fig. 4D). These results suggest that sensory innervation in sclerotic endplates is potentially related to spinal pain behavior.

**Retrograde and anterograde tracing of sensory innervation.** To demonstrate CGRP+ sensory innervation in endplates during

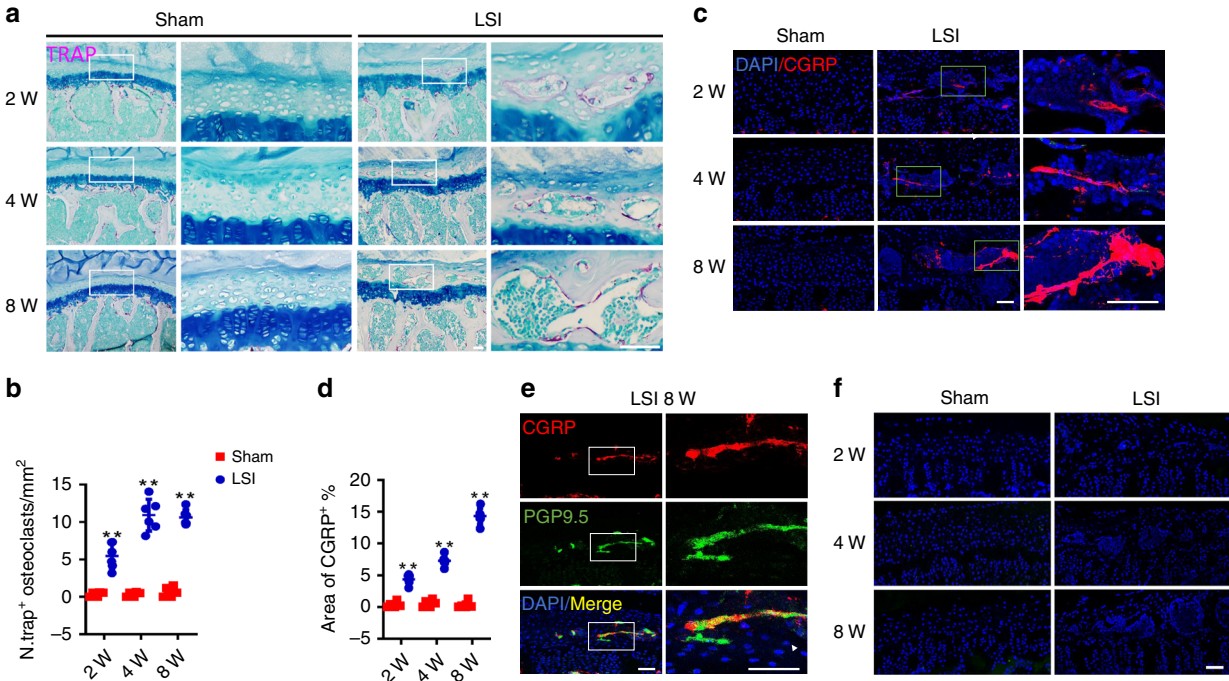

**Fig. 2 Sensory innervation in endplates correlates with increase of osteoclasts in LSI model. a** Representative images of coronal mouse caudal endplate sections of L4/5 stained for TRAP (magenta) at 2, 4, and 8 weeks after LSI or sham surgery. Scale bars, 50 μm. **b** Quantitative analysis of the number of TRAP+ cells in endplates. **c** Representative immunofluorescent images of CGRP+ sensory nerve fibers (red) and DAPI (blue) staining of nuclei in mouse caudal endplates of L4/5 at 2, 4, and 8 weeks after LSI or sham surgery. Scale bars, 50 μm. **d** Percentage of CGRP+ area in endplates. **e** Representative images of immunofluorescent analysis of CGRP+ (red), PGP9.5+ (green) nerve fibers, and DAPI (blue) staining of nuclei in mouse caudal endplates of L4/5 at 8 weeks after LSI or sham surgery. Scale bars, 50 μm. **f** Representative images of immunofluorescent analysis of IB4+ (green) sensory nerve fibers and DAPI (blue) staining of nuclei in mouse caudal endplates of L4/5 at 2, 4, and 8 weeks after LSI or sham surgery. Scale bars, 100 μm. **p < 0.01 compared with the sham surgery mice at the corresponding time points. n = 6 per group (**b**, **d**). Statistical significance was determined by multifactorial ANOVA, and all data are shown as means ± standard deviations. Source data are provided as a Source Data file.

spine degeneration, we conducted a retrograde tracing experiment in both LSI and aged mice. The red fluorescent tracer, DiI, was injected in the left part of the caudal endplates of L4/5 in mice at 8 weeks after LSI surgery (Fig. 4a). The T12–L6 dorsal root ganglions (DRGs) in both sides were harvested at 1 week after injection to calculate the number of DiI+ neurons. We observed that DiI was retrograded mainly to the left T13-L3 DRGs, especially the left L1 and L2 DRGs in LSI mice, whereas no DiI+ neurons were detected in the T12–L6 DRGs of sham surgery mice (Fig. 4b, c). Immunofluorescent staining of the DRG sections demonstrated that DiI in the left L1 and L2 DRGs was co-localized mainly with CGRP+, but not IB4+ neurons in LSI mice (Fig. 4d, e).

The anterograde-tracing experiment was performed by labeling the L1 and L2 DRG neurons in both sides with injection of DiI at 8 weeks after LSI or sham surgery. Abundant DiI-labeled sensory nerves were seen in the porous areas of endplates of L4/5 in LSI mice, but not in sham surgery mice (Fig. 4f, g). Similarly, the innervation of DiI-labeled sensory nerves in the porous areas of endplates of aged mice was also observed in the anterograde-tracing experiment (Fig. 4h, i). Taken together, these findings suggest nociceptive innervation in the sclerotic endplates of LSI and aged mice.

**PGE2/EP4 signaling mediates spinal hypersensitivity.** To elucidate the signaling mechanism of endplate sclerosis-mediated pain behavior, we examined the expression of several inflammatory cytokines in the endplates of LSI and sham surgery mice by using quantitative real-time polymerase chain reaction (qRT-PCR). We found that messenger ribonucleic acid levels of

prostaglandin E synthase (PGES), cox-2, interleukin (IL)-1β, IL-17, IL-2, and tumor necrosis factor (TNF)-α increased significantly in the lumbar endplates at 4 weeks after LSI relative to sham surgery, especially PGES (Fig. 5a). Immunostaining further confirmed a significant increase in cox-2 in the endplates at 4 and 8 weeks after LSI surgery (Fig. 5b, top). The increases of PGES, cox-2, and IL-1β can contribute to the synthesis of PGE2 in the endplates, which was validated by immunostaining (Fig. 5b, bottom) and enzyme-linked immunoabsorbent assay (ELISA) (Fig. 5c). The increase of PGE2 in endplates peaked at 4 weeks and remained high at 8 and 12 weeks after LSI surgery (Fig. 5c). To explore the potential source of PGE2 in porous endplates, the co-immunostaining for cox-2 with F4/80, cox-2 with osteocalcin (OCN), and cox-2 with TRAP were conducted, respectively. The results demonstrated that the COX-2 was co-localized with F4/80+, some OCN+, and a few TRAP+ cells (Supplementary Fig. 5). These data showed that the accumulated PGE2 in porous endplates was derived from different types of the cell. Immunostaining showed that EP4 was expressed in newly innervated CGRP+ nerve endings in the endplates of LSI mice (Fig. 5d). Notably, the proportion of CGRP+EP4+ neurons relative to CGRP+ neurons in L2 DRGs was significantly greater in LSI mice relative to sham surgery mice (Fig. 5e, f). Interestingly, the sodium channel Na$_v$ 1.8 was also expressed in newly innervated CGRP+ nerve ending in the endplates of LSI mice, as demonstrated by immunostaining (Fig. 5g). Moreover, the proportion of CGRP+Na$_v$1.8+ neurons relative to CGRP+ neurons in L2 DRGs increased significantly at 4 and 8 weeks after LSI surgery (Fig. 5h, i).

To examine the potential role of PGE2/EP4 in pain transduction, we generated a sensory neuron specific *EP4*

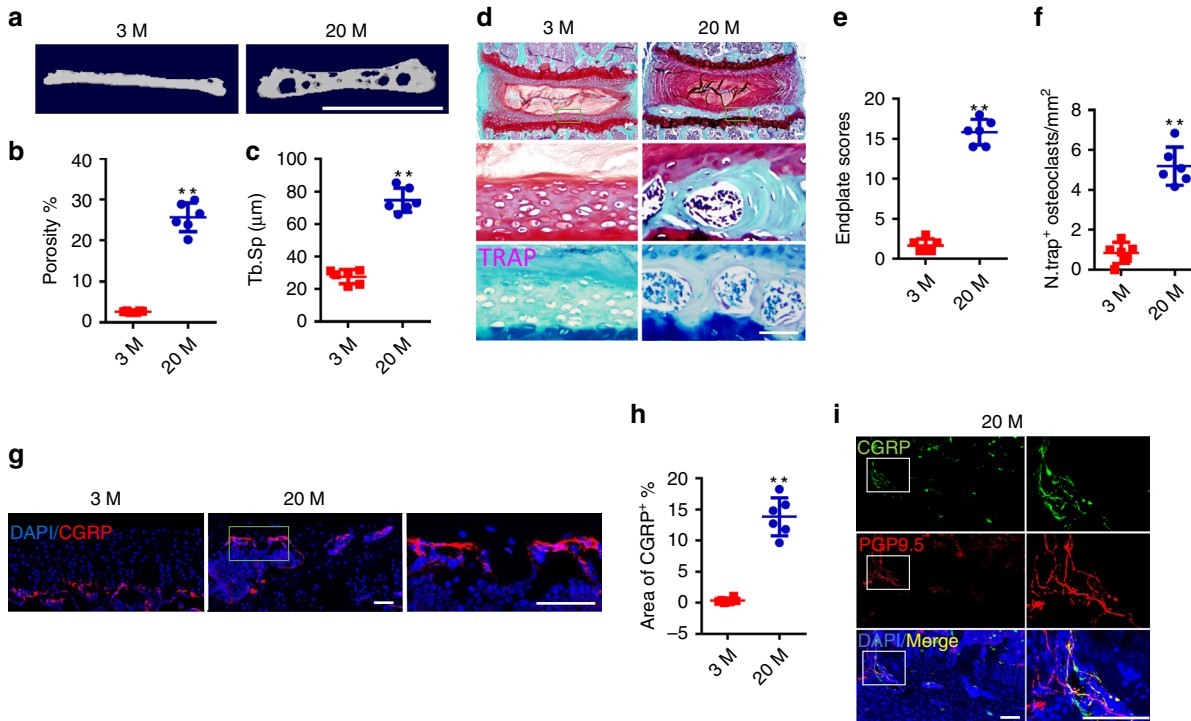

**Fig. 3 Sensory innervation in endplates correlates with increase of osteoclasts during aging. a** Representative three-dimensional high-resolution μCT images of the mouse caudal endplates of L4/5 (coronal view) in 3-month-old and 20-month-old mice. Scale bars, 1 mm. **b, c** Quantitative analysis of the total porosity (**b**) and trabecular separation (Tb. Sp; **c**) of the mouse caudal endplates of L4/5 determined by μCT. **d** Top and middle, representative images of safranin O and fast green staining of coronal caudal endplate sections of L4/5 in 3-month-old and 20-month-old mice, proteoglycan (red) and cavities (green). Bottom, representative images of TRAP (magenta) staining of coronal sections of the caudal endplates of L4/5 in 3-month-old and 20-month-old mice. Scale bars, 50 μm. **e** Endplate scores in 3-month-old and 20-month-old mice as an indication of endplate degeneration based on safranin O and fast green staining. **f** Quantitative analysis of the number of TRAP+ cells in endplates. **g** Representative images of immunofluorescent analysis of CGRP+ sensory nerve fibers (red) and DAPI (blue) staining of nuclei in mouse caudal endplates of L4/5 in 20-month-old and 3-month-old mice. Scale bars, 50 μm. **h** Quantitative analysis of the percentage of CGRP+ area in endplates. **i** Representative images of immunofluorescent analysis of CGRP+ (green), PGP9.5+ (red) nerve fibers, and DAPI (blue) staining of nuclei in mouse caudal endplates of L4/5 in 20-month-old mice. Scale bars, 50 μm. $*p < 0.05$, $**p < 0.01$ compared with the 3-month-old mice. $n = 6$ per group (**b, c, e, f,** and **h**). Statistical significance was determined by two-tailed Student's $t$ test, and all data are shown as means ± standard deviations. Source data are provided as a Source Data file.

knockout mice ($Avil^{Cre}$; $EP4^{flox/flox}$, named $EP4^{-/-}$ mice). TRAP staining demonstrated that there was no significant difference in the number of TRAP+ osteoclasts in endplates between $EP4^{f/f}$ and $EP4^{-/-}$ mice of sham surgery group or LSI surgery group (Supplementary Fig. 6A, B). Asante NaTRIUM Green 2 acetoxymethyl (ANG-2 AM), a sodium indicator, was loaded into the DRG neurons to detect the real-time sodium influx. Interestingly, PGE2 significantly stimulated the enhancement of the fluorescent intensity in neurons (Fig. 6a, left and 6b), indicating increased sodium influx. Importantly, this effect was abolished in the DRG neurons of $EP4^{-/-}$ mice (Fig. 6a, right and 6c). To determine the mechanism by which PGE2 induces sodium influx, we examined whether PGE2 can activate the cyclic adenosine monophosphate (cAMP) pathway in sensory neurons. Western blot and fluorescent staining demonstrated that PGE2-induced cAMP production activates protein kinase A (PKA) and cAMP response element binding (CREB) protein, and the activation was abrogated by PKA inhibitor or $EP4^{-/-}$ (Fig. 6d–f). Furthermore, PGE2-induced sodium influx was ablated by PKA inhibitor or small interfering ribonucleic acid (siRNA) for $Na_v$ 1.8 (Fig. 6g, first to third columns and 6h–j), and cAMP rescued sodium influx in $EP4^{-/-}$ mice (Fig. 6g, fourth column and 6k). These results demonstrate that PGE2 activates EP4 in sensory neurons to induce sodium influx of $Na_v$ 1.8 through cAMP signaling with implications for pain transduction.

To evaluate whether the PGE2/EP4 pathway in sensory fibers is associated with spinal pain behavior, we performed pain behavior tests, including pressure tolerance, spontaneous activity, and von Frey analysis. The threshold of pressure tolerance in response to pressure stimulation was lower in $EP4^{-/-}$ mice relative to $EP4^{f/f}$ mice after LSI surgery (Fig. 7a). Analysis of spontaneous activity revealed that the distance traveled, maximum speed, mean speed, and activity time per 24 h were significantly preserved in $EP4^{-/-}$ mice relative to $EP4^{f/f}$ mice after LSI surgery (Fig. 7b–e). Consistently, mechanical hyperalgesia of the hind paw was also ameliorated in $EP4^{-/-}$ mice relative to $EP4^{f/f}$ mice after LSI surgery (Fig. 7f, g).

**Decreased osteoclast attenuates sensory innervation and pain.** To evaluate whether CGRP+ sensory innervation in the end-plates is induced by osteoclasts, we bred *Dmp1-Cre* mice with osteocyte-derived receptor activator of nuclear factor kappa-B ligand (*Rankl*) floxed mice to generate $Dmp1^{Cre}$; $Rankl^{flox/flox}$ mice (named $Rankl^{-/-}$ mice) to knockout *Rankl* specifically in DMP1+ osteocytes. Deficiency of *Rankl* in osteocytes leads to a decrease in osteoclast number and a severe osteopetrotic phenotype[41,42]. The trabecular BV/TV, Tb.N, and Tb.Th increased and Tb.Sp decreased significantly in RANKL$^{-/-}$ mice relative to RANKL$^{f/f}$ mice in μCT analysis (Supplementary Fig. 7A–F), indicating the osteopetrotic vertebrae in $Rankl^{-/-}$

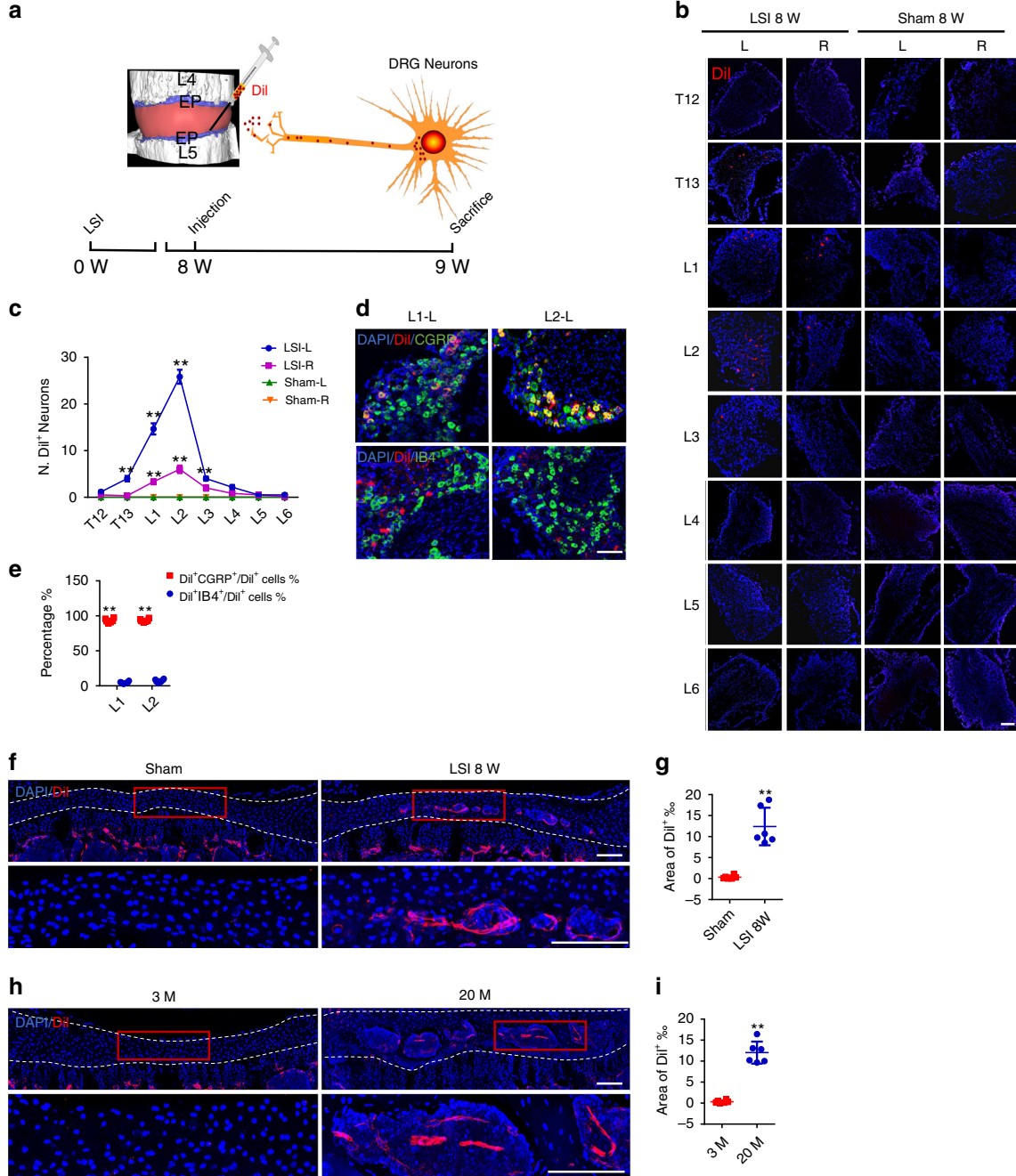

**Fig. 4 Sensory innervation in endplates is validated by retrograde and anterograde tracing. a** Model of retrograde tracing of the sensory innervation in the endplates of L4/5. The T12–L6 DRGs were harvested at 1 week after injection of DiI in the left part of the mouse caudal endplates at 8 weeks after LSI or sham surgery. **b** Representative images of DiI+ (red) sensory neurons and DAPI (blue) staining of nuclei in the left (L) and right (R) side DRGs. Scale bars, 200 μm. **c** Quantitative analysis of the number of DiI+ cells of (**b**). **p < 0.01 compared with the sham surgery mice at the corresponding side. n = 6 per group. **d** Top, representative images of DiI+ (red) and CGRP+ sensory neurons and DAPI (blue) staining of nuclei in the left (L) side DRGs of L1 and L2. Bottom, representative images of DiI+ (red) and IB4+ sensory neurons and DAPI (blue) staining of nuclei in the left (L) side DRGs of L1 and L2. Scale bars, 100 μm. **e** Quantitative analysis of (**d**). **p < 0.01 compared with the percentage of DiI+IB4+ cells to DiI+ cells in the corresponding DRG. n = 6 per group. **f–h** The anterograde-tracing analysis of L1 and L2 DRG neuronal fibers innervation into the caudal endplates of L4/5. The DiI was injected in the L1 and L2 DRGs at 8 weeks after LSI and sham surgery; or in 3-month-old and 20-month-old mice. Representative images of DiI+ (red) sensory nerve fibers in the caudal endplates of L4/5 in LSI and sham surgery mice (**f**) or in 3-month-old and 20-month-old group (**h**) at 1 week after injection. Scale bars, 100 μm. Quantitative analysis of the percentage of DiI+ area in endplates in LSI and sham surgery group (**g**) or in 3-month-old and 20-month-old group (**i**). **p < 0.01 compared with the 3-month-old mice. n = 6 per group (**g**, **i**). Statistical significance was determined by multifactorial ANOVA, and all data are shown as means ± standard deviations. Source data are provided as a Source Data file.

mice. The sclerosis of endplates was delayed significantly in *Rankl*$^{-/-}$ mice relative to their age-matched wild-type (WT) littermates (*Rankl*$^{f/f}$ mice) after LSI surgery, as shown by decreases in the porosity and trabecular separation of endplates by μCT analysis

(Fig. 8a–c). The delayed sclerosis of endplates in *Rankl*$^{-/-}$ mice was further validated by Safranin O and fast green staining, with less porous areas and significant lower endplate scores (Fig. 8d, e). The number of TRAP+ osteoclasts decreased significantly in endplates at

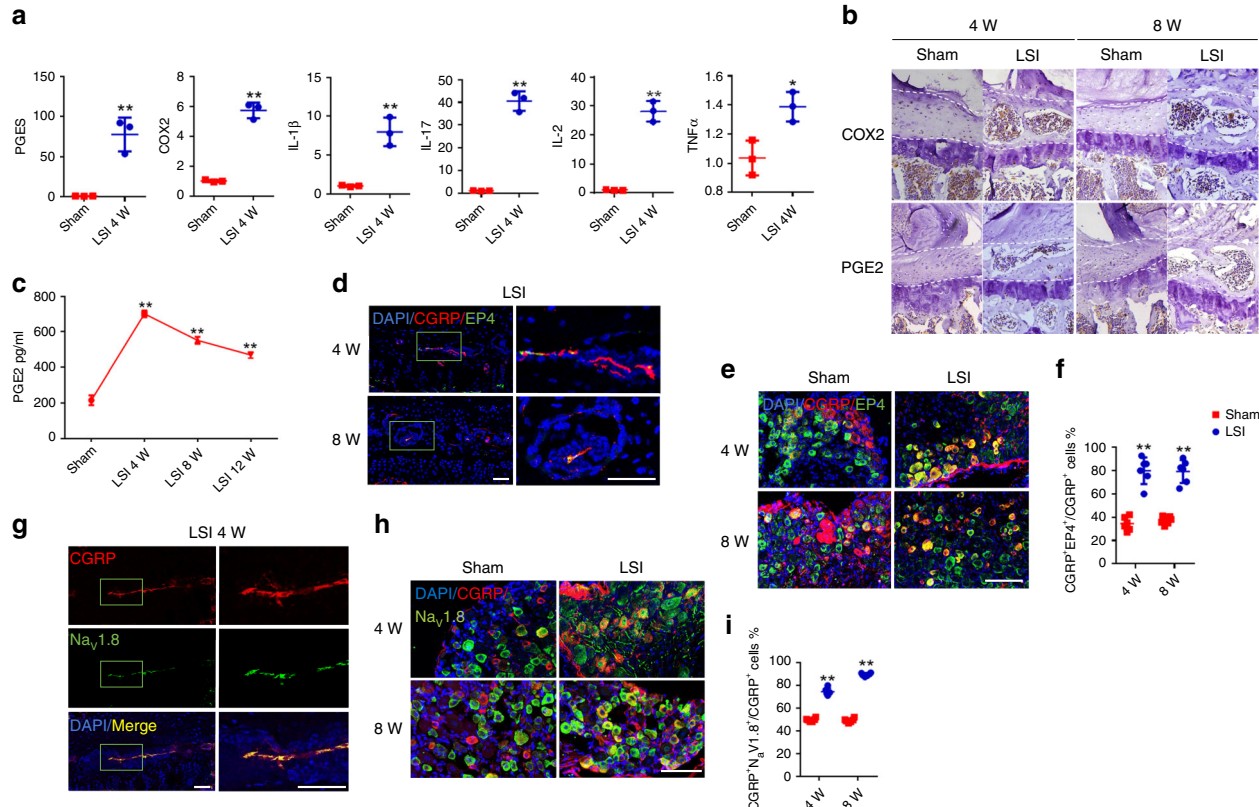

**Fig. 5 PGE2/EP4 contributes to the spinal pain hypersensitivity. a** Quantitative analysis of the expression of PGE synthetase (PGES), cox-2, IL-1β, IL-17, IL-2, and TNF-α in lumbar endplates at 4 weeks after LSI determined by qRT-PCR. **b** Representative images of Immunohistochemical analysis of Cox-2 (brown; top) or PGE2 (brown; bottom) in the caudal endplates of L4/5 at 4 and 8 weeks after LSI or sham surgery. Scale bars, 50 μm. **c** ELISA analysis of PGE2 concentration in the lysate of lumbar endplates at 4, 8, and 12 weeks after LSI surgery. *$p < 0.05$, **$p < 0.01$ compared with the sham surgery mice. $n = 3$ (**a, c**). **d** Representative images of immunofluorescent analysis of CGRP (red), EP4 (green) staining, and DAPI (blue) staining of nuclei in the caudal endplates of L4/5 at 4 and 8 weeks after LSI surgery. Scale bars, 50 μm. **e** Representative images of immunofluorescent analysis of CGRP (red), EP4 (green) staining, and DAPI (blue) staining of nuclei in the L2 DRGs at 4 and 8 weeks after LSI surgery. Scale bars, 100 μm. **f** Quantitative analysis of percentage of CGRP+ EP4+ cells to CGRP+ cells in the L2 DRGs at 4 and 8 weeks after LSI surgery. **g** Representative images of immunofluorescent analysis of CGRP (red), Na$_v$ 1.8 (green) staining, and DAPI (blue) staining of nuclei in the caudal endplates of L4/5 at 4 weeks after LSI surgery. Scale bars, 50 μm. **h** Representative images of immunofluorescent analysis of CGRP (red), Na$_v$ 1.8 (green) staining, and DAPI (blue) staining of nuclei in the L2 DRGs at 4 and 8 weeks after LSI surgery. Scale bars, 100 μm. **i** Quantitative analysis of percentage of CGRP+ Na$_v$ 1.8+ cells to CGRP+ cells in the L2 DRGs at 4 and 8 weeks after LSI surgery. **$p < 0.01$ compared with the sham surgery mice at the corresponding time points. $n = 6$ per group (**f, i**). Statistical significance was determined by multifactorial ANOVA, and all data are shown as means ± standard deviations. Source data are provided as a Source Data file.

4 and 8 weeks after LSI surgery in Rankl−/− mice relative to Rankl^f/f mice (Fig. 8f, g). Notably, immunostaining showed that instability-induced CGRP+ sensory innervation in the endplates was inhibited in Rankl−/− mice (Fig. 8h, i). Although the density of CGRP+ sensory nerve fibers increased slightly at 8 weeks after LSI surgery in Rankl−/− mice, it was still significantly lower than that of Rankl^f/f mice (Fig. 8h, i), indicating that osteoclast activity was associated with CGRP+ sensory innervation in the endplates. Moreover, the sprouting of CD31+EMCN+ blood vessels in the endplates was also significantly inhibited in Rankl−/− mice relative to Rankl^f/f mice (Supplementary Fig. 8A, B).

To examine whether osteoclast activity-induced sensory innervation in endplates is associated with spinal pain behavior, we performed pain behavior tests, including pressure tolerance, spontaneous activity, and von Frey analysis. The threshold of pressure tolerance was significantly lower in Rankl−/− mice relative to Rankl^f/f mice after LSI surgery (Fig. 8j). Analysis of spontaneous activity revealed that the distance traveled, maximum speed, mean speed, and activity time per 24 h were significantly preserved in Rankl−/− mice relative to Rankl^f/f mice after LSI surgery (Fig. 8k–n). Consistently, mechanical hyperalgesia of the hind paw was also ameliorated in Rankl−/− mice relative to Rankl^f/f mice after LSI surgery (Fig. 8o, p). Nerve innervation in the annulus fibrosus (AF) has been implicated in back pain development[43]. CGRP+ sensory innervation was observed in the AF of LSI mice. There is no significant difference in the density of newly innervated sensory nerves between Rankl−/− mice and Rankl^f/f mice (Supplementary Fig. 9A, B). Taken together, these results indicate that sensory innervation in vertebral endplates is associated with osteoclast activity and pain behavior.

**Knockout of Netrin-1 abrogates sensory innervation and pain.** Netrin-1 is an important axon guidance factor to attract nerve protrusion[44–46]. We previously reported that osteoclasts can secret Netrin-1 to attract sensory nerve growth[47]. Immunostaining demonstrated that TRAP+ osteoclasts were the source of Netrin-1 in sclerotic endplates after LSI surgery (Fig. 9a). The level of Netrin-1 in lower lumbar endplates increased gradually at 4, 8, and 12 weeks after LSI surgery relative to sham surgery mice, as indicated by ELISA (Fig. 9b). Deleted in colorectal cancer (DCC) is identified as the receptor that mediates Netrin-1-induced neuronal sprouting[48,49]. Immunostaining revealed that DCC was co-localized with CGRP+ sensory nerve fibers in endplates after LSI surgery (Fig. 9c).

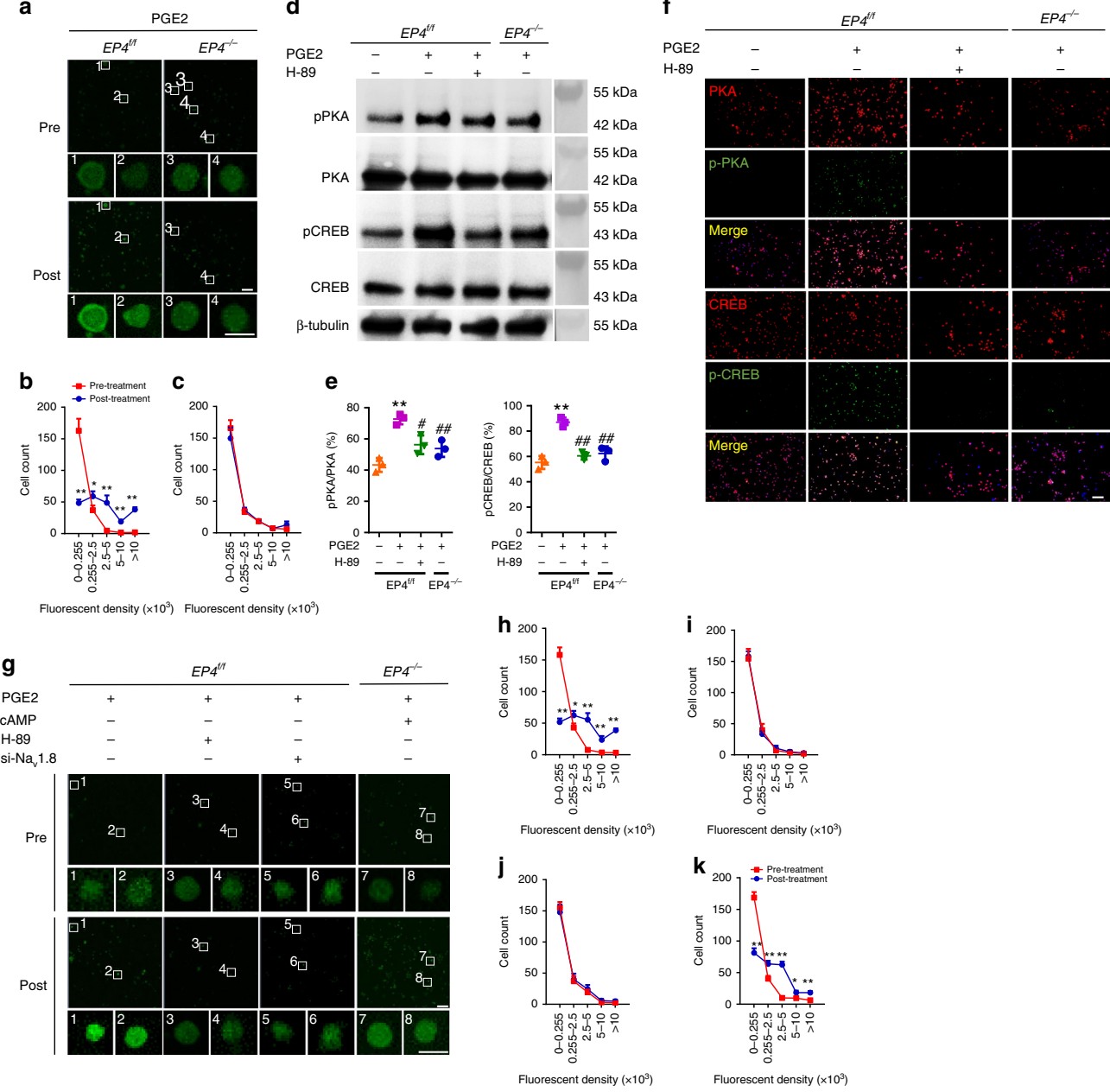

**Fig. 6 PGE2 stimulates PKA/CREB signaling through EP4 to induce sodium influx. a** Representative images of sodium indicator (green) analysis pre- and post-PGE2 (20 μM) stimulation for 5 min in primary DRG neurons from *EP4 $^{f/f}$* or *EP4 $^{-/-}$* mice, indicating sodium influx. Scale bar, 100 μm. Magnification, scale bar, 20 μm. **b, c** Quantitative analysis of the fluorescent density distribution of the 1st (**b**) and 2nd (**c**) column in (**a**). *$p < 0.05$, **$p < 0.01$ compared with the corresponding pre-treatment group. $n = 3$ per group. **d** Western blots of the phosphorylation of PKA and CREB in primary DRG neurons treated with PGE2 (20 μM) for 30 min and PKA inhibitor (H-89, 10 μM) for 60 min. **e** Quantitative analysis of (**d**). **$p < 0.01$ compared with the negative control group from EP4$^{f/f}$ mice. #$p < 0.05$, ##$p < 0.01$ compared with only PGE2 treatment group from EP4$^{f/f}$ mice $n = 3$ per group. **f** First to third row, representative images of immunofluorescent analysis of PKA (red), p-PKA (green) staining, and DAPI (blue) staining of nuclei; 4th to 6th row, representative images of immunofluorescent analysis of CREB (red), p-CREB (green) staining, and DAPI (blue) staining of nuclei pre- and post-PGE2 (20 μM) stimulation combined with H-89 (10 μM) in primary DRG neurons from *EP4 $^{f/f}$* or *EP4 $^{-/-}$* mice. Scale bar, 100 μm. **g** Representative images of sodium indicator (green) analysis pre- and post-PGE2 (20 μM) stimulation combined with cAMP, PKA inhibitor (H-89), or siRNA for Na$_v$ 1.8 (si-Na$_v$1.8) in primary DRG neurons from *EP4 $^{f/f}$* or *EP4 $^{-/-}$* mice. Scale bar, 100 μm. Magnification, scale bar, 20 μm. **h–k** Quantitative analysis of the fluorescent density distribution of the 1st (**h**), 2nd (**i**), 3rd (**j**), 4th (**k**) column in (**g**). *$p < 0.05$, **$p < 0.01$ compared with the corresponding pre-treatment group. $n = 3$ per group. Statistical significance was determined by multifactorial ANOVA, and all data are shown as means ± standard deviations. Source data are provided as a Source Data file.

To determine whether osteoclast-derived Netrin-1 is responsible for sensory innervation, we cross-bred *Trap-Cre* mice with *Netrin-1* floxed mice to knockout *Netrin-1* in TRAP$^+$ lineage cells (*Trap$^{Cre}$*; *Netrin-1$^{flox/flox}$*, named *Netrin-1$^{-/-}$* mice). Safranin O and fast green staining demonstrated that the instability-induced

sclerosis of endplates was not different in *Netrin-1$^{-/-}$* mice compared with their age-matched littermates (herein, *Netrin-1$^{f/f}$* mice) (Fig. 9d), as evidenced by endplate scores (Fig. 9e). TRAP staining showed similar numbers of osteoclasts in the endplates of *Netrin-1$^{-/-}$* mice and *Netrin-1$^{f/f}$* mice after LSI surgery (Fig. 9f,

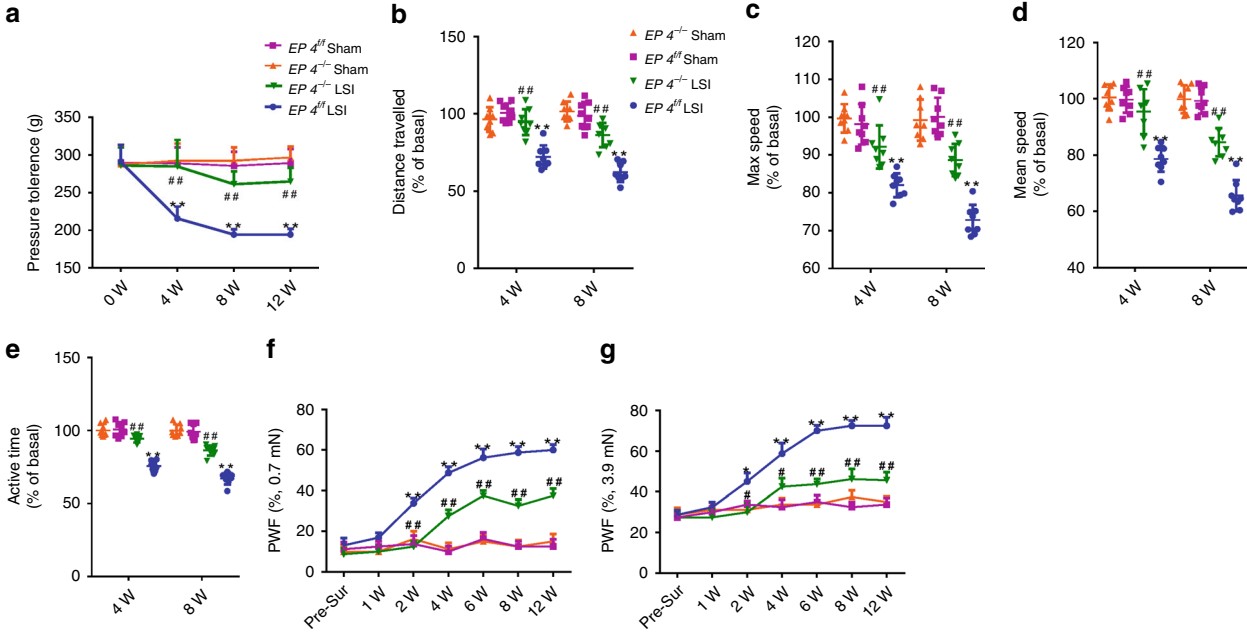

**Fig. 7 EP4 knockout in sensory nerve attenuates spinal pain behavior. a–g** Quantitative analysis of spinal pain-related behavior tests, including pressure hyperalgesia (**a**), spontaneous distance traveled (**b**), maximum speed (**c**), mean speed (**d**), active time (**e**) per 24 h, and hind paw withdrawal frequency responding to mechanical stimulation (0.7 mN; f and 3.9 mN; **g**) in $EP4^{-/-}$ or $EP4^{f/f}$ mice overtime after LSI or sham surgery. $*p < 0.05$, $**p < 0.01$ compared with $EP4^{f/f}$ sham surgery mice, $\#p < 0.05$, $\#\#p < 0.01$ compared with $EP4^{f/f}$ LSI surgery mice at the corresponding time points. $n = 8$ per group (**a–g**). Statistical significance was determined by multifactorial ANOVA, and all data are shown as means ± standard deviations. PWF paw withdraw frequency. Source data are provided as a Source Data file.

g), suggesting that Netrin-1 is not involved in endplate sclerosis. The density of CGRP$^+$ nociceptive nerves did not increase in $Netrin-1^{-/-}$ mice, although the number of TRAP$^+$ osteoclasts significantly increased in the endplates of $Netrin-1^{-/-}$ mice after LSI surgery (Fig. 9h, i). Moreover, the sprouting of CD31$^+$EMCN$^+$ blood vessels in the endplates was significantly inhibited in $Netrin-1^{-/-}$ mice relative to $Netrin-1^{f/f}$ mice (Supplementary Fig. 10A, B).

Pain behavior tests demonstrated that pressure hyperalgesia (Fig. 9j), loss of spontaneous activity (Fig. 9k–n), and mechanical hyperalgesia of the hind paw (Fig. 9o, p) were significantly lower at 4 and 8 weeks after LSI surgery in $Netrin-1^{-/-}$ mice relative to Netrin-1$^{f/f}$ mice. As in $Rankl^{-/-}$ mice, CGRP$^+$ sensory innervation in the AF did not decrease in Netrin-1$^{-/-}$ mice (Supplementary Fig. 11A, B). Together, these results indicate that osteoclast-derived Netrin-1 mediates sensory innervation in vertebral endplates responsible for spinal pain behavior.

## Discussion

LBP is difficult to diagnose and treat because of limited knowledge about its source. Current treatments, including activity modification, physical therapy, and pharmaceutical agents aim to alleviate the pain, but not to modify the disease[9,50]. Surgical treatment, such as disc replacement and lumbar fusion, are the most common final treatments. Efforts to understand the causes of LBP have focused largely on sensory innervation in the degenerative IVD[51]. However, IVD degeneration is frequently asymptomatic. Importantly, endplates undergo sclerosis during aging and become porous, which is clinically associated with LBP[52–54]. We have shown that aberrant mechanical loading induces sclerosis of the endplates with elevated osteoclast activity and activates excessive TGF-β1 to recruit mesenchymal stromal cells[20]. Here, we showed elevated level of PGE2 and sensory innervation in the porous sclerotic endplates. PGE2 could activate

its receptor EP4 in the newly innervated sensory nerves to cause spinal pain. We recently reported that sensory nerves can monitor bone-density changes through the concentration of PGE2. Osteoblast-secreted PGE2 activates its receptor EP4 in the sensory nerves to tune down sympathetic tone for osteoblastic bone formation[30]. The porosity of the sclerotic endplates resembles low bone mineral density to promote PGE2 concentration and sensory innervation causing pain. These indicate that PGE2 is a key mediator of pain hypersensitivity and endplate sclerosis. Our findings suggest that inhibition of endplate sclerosis to reduce sensory innervation or blocking the PGE2/EP4 pathway could ameliorate spinal pain behavior.

Sensory nerves and CD31$^+$EMCN$^+$ vessels appeared largely in the porous areas of the sclerotic endplates. Osteoclast resorption likely causes the porosity of sclerotic endplates. Osteoclastic lineage cells secrete Netrin-1 to induce CGRP$^+$ sensory nerve fiber extrusion and innervation in the space generated by osteoclast resorption. In addition, Netrin-1 is suggested to be a potential pro-angiogenic factor that promotes endothelial cell migration and capillary-like tube formation[55,56]. We have shown that pre-osteoclasts secrete platelet-derived growth factor-BB to induce angiogenesis coupled with osteogenesis during bone formation[57]. Thus, osteoclast activity in the sclerotic endplates is the primary cause of sensory innervation and angiogenesis for LBP. Indeed, a decreased number of osteoclasts in $Rankl^{-/-}$ mice led to significantly decreased sensory innervation and angiogenesis in the endplates. Moreover, the density of sensory innervation and angiogenesis was reduced significantly by conditional knockout of $Netrin-1$ in TRAP$^+$ osteoclastic cells. Therefore, osteoclast lineage cells instigate porosity of sclerotic endplates and spinal pain behavior.

The IVD and the endplate act in as a functional unit in the spine. Our previous study revealed that aberrant mechanical loading induced hypertrophy of chondrocytes and calcification of the endplates, leading to an osteoclast-initiated remodeling

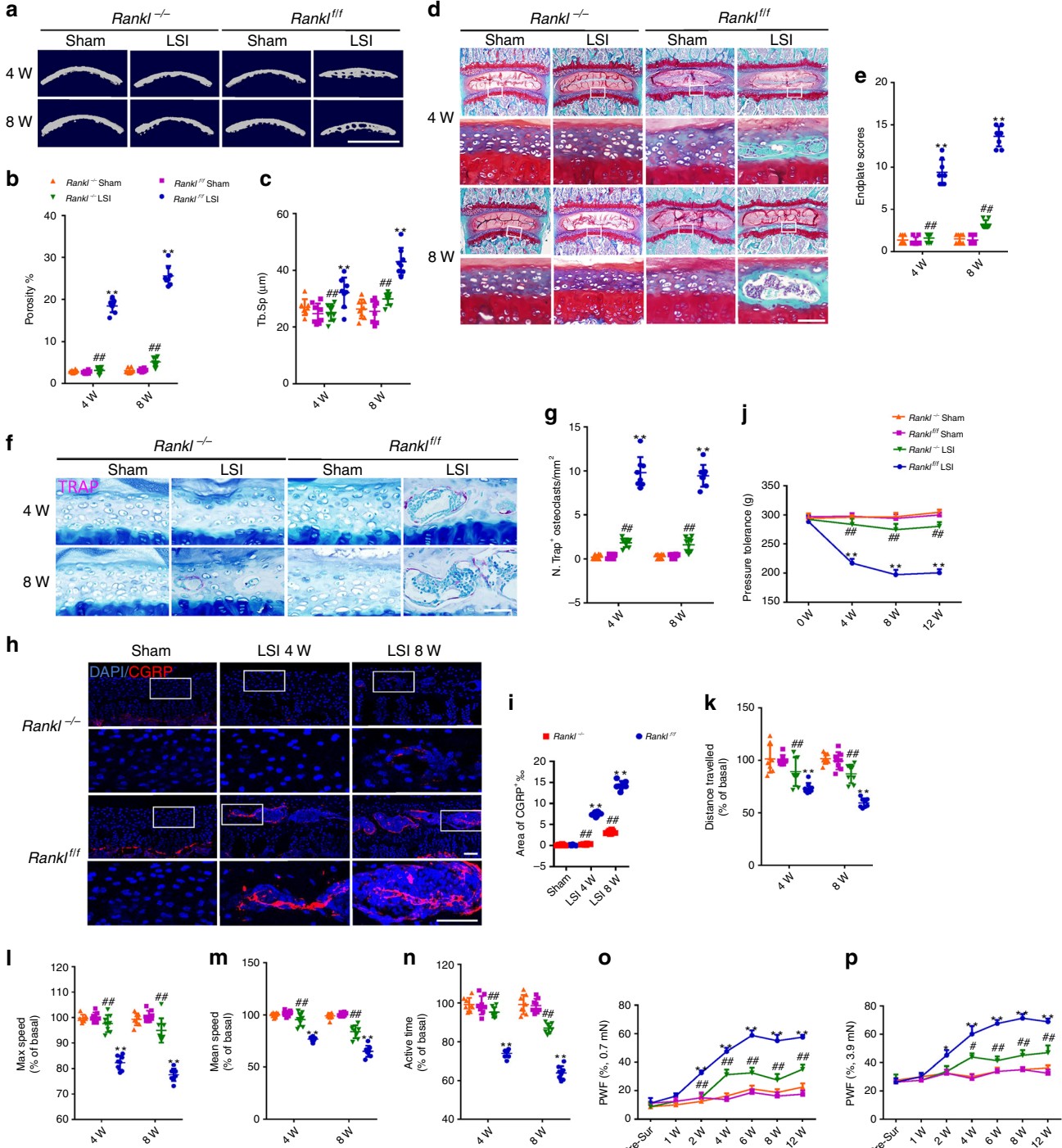

**Fig. 8 Decreased osteoclasts activity diminishes sensory innervation and attenuates pain. a** Representative μCT images of the caudal endplates of L4/5 (coronal view) in *Rankl*[−/−] or *Rankl*[f/f] mice at 4 and 8 weeks after LSI or sham surgery. Scale bars, 1 mm. **b, c** Quantitative analysis of the total porosity (**b**) and trabecular separation (Tb. Sp; **c**) of the mouse caudal endplates of L4/5 determined by μCT. **d** Representative images of safranin O and fast green staining of coronal sections of the caudal endplates of L4/5 in *Rankl*[−/−] or *Rankl*[f/f] mice at 4 and 8 weeks after LSI or sham surgery. Scale bars, 50 μm. **e** Endplate scores of the caudal endplates. **f** Representative images of TRAP (magenta) staining of coronal sections of the caudal endplates of L4/5 in *Rankl*[−/−] or *Rankl*[f/f] mice at 4 and 8 weeks after LSI or sham surgery. Scale bars, 50 μm. **g** Quantitative analysis of the number of TRAP+ cells in the caudal endplates. **h** Representative images of immunofluorescent analysis of CGRP+ sensory nerve fibers (red) and DAPI (blue) staining of nuclei in caudal endplates of L4/5 in *Rankl*[−/−] or *Rankl*[f/f] mice at 4 and 8 weeks after LSI or sham surgery. Scale bars, 50 μm. **i** Quantitative analysis of the percentage of CGRP+ area in caudal endplates. **j-p** Quantitative analysis of spine pain-related behavior tests, including pressure hyperalgesia (**j**), distance traveled (**k**), maximum speed (**l**), mean speed (**m**), active time (**n**) per 24 h, and hind paw withdrawal frequency responding to mechanical stimulation (0.7 mN; **o** and 3.9 mN; **p**) in *Rankl*[−/−] or *Rankl*[f/f] mice overtime after LSI or sham surgery. **p < 0.01 compared with *Rankl*[f/f] sham surgery mice, ##p < 0.01 compared with *Rankl*[f/f] LSI surgery mice at the corresponding time points. n = 8 per group (**b, c, e, g**, and **i–p**). Statistical significance was determined by multifactorial ANOVA, and all data are shown as means ± standard deviations. Source data are provided as a Source Data file.

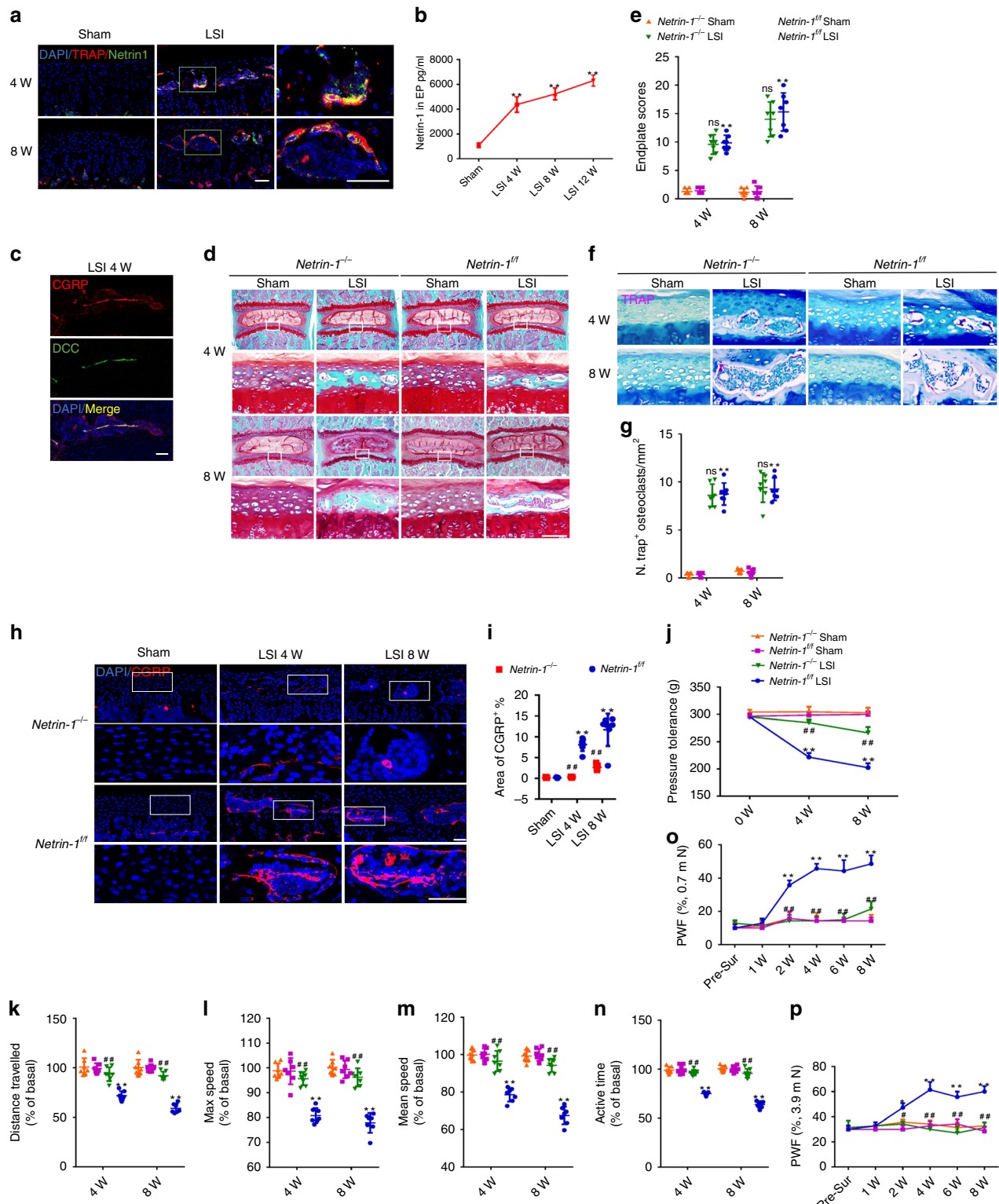

sclerotic process. As a result, the expansion of mineralized endplates narrowed the intervertebral space and generated pathological static compression on the intervertebral disc, leading to its degeneration[20]. The AF has been suggested as the main source of discogenic pain. In the physiological condition, only the outer third of the AF is innervated by sensory nerves, whereas in degenerative discs, the nerve can grow into the middle third, or even the inner third of the AF[43]. In this study, we observed innervation of CGRP+ sensory fibers into the AF in WT mice after LSI surgery, but the LSI-

induced newly innervated sensory nerves in the AF were not decreased in Rankl−/− and Netrin-1−/− mice. On the other hand, during sclerosis of the endplates, osteoclast resorption generates porous areas and abundant sensory innervation along with angiogenesis. Importantly, the density of CGRP+ sensory nerve fibers in endplates and spinal pain behavior were significantly ameliorated in Rankl−/− and Netrin-1−/− mice after LSI surgery. These results indicate that osteoclast-induced sensory innervation in the sclerotic endplates is the primary source of nociceptors for spinal pain.

**Fig. 9 Knockout of netrin-1 abrogates sensory innervation and spinal pain. a** Representative images of immunofluorescent analysis of TRAP$^+$ (red), Netrin-1$^+$ (green), and DAPI (blue) staining of nuclei in caudal endplates of L4/5 after LSI or sham surgery. Scale bars, 50 μm. **b** ELISA analysis of Netrin-1 concentration in the lysate of lumbar endplates after LSI surgery. $**p < 0.01$ compared with the sham surgery mice. $n = 3$ per group. **c** Representative images of immunofluorescent analysis of CGRP$^+$ (red), DCC$^+$ (green), and DAPI (blue) staining of nuclei in caudal endplates of L4/5 after LSI surgery. Scale bars, 50 μm. **d** Representative images of safranin O and fast green staining of the caudal endplates of L4/5 in Netrin-1$^{-/-}$ or Netrin-1$^{f/f}$ mice after LSI or sham surgery. Scale bars, 50 μm. **e** Endplate scores of the caudal endplates. **f** Representative images of TRAP (magenta) staining of the caudal endplates of L4/5 in Netrin-1$^{-/-}$ or Netrin-1$^{f/f}$ mice after LSI or sham surgery. Scale bars, 50 μm. **g** Quantitative analysis of (**f**). **h** Representative images of immunofluorescent analysis of CGRP$^+$ (red) and DAPI (blue) staining of nuclei in caudal endplates of L4/5 in Netrin-1$^{-/-}$ or Netrin-1$^{f/f}$ mice after LSI or sham surgery. Scale bars, 50 μm. **i** Quantitative analysis of (**h**). **j–p** Quantitative analysis of spinal pain-related behavior tests, including pressure hypersensitivity (**j**), distance traveled (**k**), maximum speed (**l**), mean speed (**m**), active time (**n**) per 24 h, and hind paw withdrawal frequency responding to mechanical stimulation (0.7 mN; **o** and 3.9 mN; **p**) in Netrin-1$^{-/-}$ or Netrin-1$^{f/f}$ mice after LSI or sham surgery. $*p < 0.05$, $**p < 0.01$ compared with Netrin-1$^{f/f}$ sham surgery mice, $^\#p < 0.05$, $^{\#\#}p < 0.01$ compared with Netrin-1$^{f/f}$ LSI surgery mice at the corresponding time points, ns, no significant difference, compared with Netrin-1$^{f/f}$ LSI surgery mice at the corresponding time points. $n = 7$ per group (**g**, **i–p**). Statistical significance was determined by multifactorial ANOVA, and all data are shown as means ± standard deviations. Source data are provided as a Source Data file.

The sign of the hind paw mechanical allodynia is considered as the secondary indicator of spinal pain-associated behaviors. Several studies reported that the hind paw mechanical allodynia develops in LBP animal models as the secondary hypersensitivity[58–61]. Among these works of literature, one study about the lumbar facet joint osteoarthritis-induced spinal pain excluded the local inflammation or nerve injury (with negative straight leg-raising test)[61]. Our data also show the development of hind paw mechanical allodynia in the LSI model. One study demonstrated that the mouse sciatic nerve predominantly origins from the L3 and L4 DRG by injecting retrograde labeling in the hind paw[62]. Our retrograde tracing data demonstrated that L3 DRG is also the partial origin of sensory nerves in the endplates of L4/5 in LSI mice. In addition, the dorsal horn of spinal cord receives inputs from several segmental DRGs[63,64]. The major monosynaptic input for the dorsal horn neurons in L4 segment is from the L4–L6 DRGs, the dorsal horn neurons in L3 segment is from the L2–L5 DRGs[65]. These anatomical features might be the basis of the hind paw mechanical hypersensitivity in the LSI model.

## Methods

### Human subjects
Human endplate samples were obtained from patients undergoing spinal fusion surgery in the Department of Spine Surgery at Xiangya Hospital (Changsha, China). Ethics committee approvals and patient consent were obtained before harvesting human tissue samples. Detailed information about the patients and groups is provided in Supplementary Table 1.

### Mice
We purchased C57BL/6J (WT) male mice from Charles River Laboratories (Wilmington, MA). We anesthetized the mice at 2 months of age with ketamine (Vetalar, Ketaset, Ketalar; 100 mg/kg, intraperitoneally) and xylazine (Rompun, Sedazine, AnaSed; 10 mg/kg, intraperitoneally). Then, the LSI model was created by resecting the L3–L5 spinous processes, and the supraspinous and interspinous ligaments to induce instability of the lumbar spine[20,39,40]. Sham operations were performed by only detachment of the posterior paravertebral muscles from L3–L5 on a separate group of mice. For the time-course experiments, mice were euthanized with an overdose of isoflurane (Forane, Baxter) inhalation at 2, 4, 8, or 12 weeks after LSI or sham surgery (10–12 per group). For the aging-induced endplate degeneration model, 3- and 20-month-old C57BL/6J (WT) male mice were purchased from Jackson Laboratory (10–12 per group).

The Avil-Cre mouse strain was provided by Dr. Xinzhong Dong (The Johns Hopkins School of Medicine, Baltimore, MD). The EP4$^{flox/flox}$ mouse strain was obtained from Dr. Brian L. Kelsall (National Institutes of Health, Bethesda, MD). The Trap-Cre mouse strain was obtained from Dr. J. J. Windle (Virginia Commonwealth University, Richmond, VA). The Netrin-1$^{flox/flox}$ mouse strain was provided by Dr. H. K. Eltzschig (University of Texas Health Science Center at Houston, Houston, TX). Dmp1-Cre and Rankl$^{flox/flox}$ mouse strains were purchased from the Jackson Laboratory (Bar Harbor, ME).

Heterozygous male Avil-Cre mice were crossed with EP4$^{flox/flox}$ mice. The offspring were intercrossed to generate the following genotypes: WT, Avil-Cre (mice expressing Cre recombinase driven by Advillin promoter), EP4$^{flox/flox}$ (mice homozygous for the EP4 flox allele, referred to as EP4$^{f/f}$ herein) and Avil-Cre; EP4$^{flox/flox}$ (conditional deletion of EP4 in Advillin lineage cells, referred to as EP4$^{-/-}$ herein). Heterozygous Dmp1-Cre mice were crossed with Rankl$^{flox/flox}$; the offspring were intercrossed to generate the following genotypes: WT, Dmp1-Cre (mice expressing Cre recombinase driven by Dmp1 promoter), Rankl$^{flox/flox}$ (mice homozygous for the Rankl flox allele, referred to as Rankl$^{f/f}$ herein), Dmp1-Cre; Rankl$^{flox/flox}$ (conditional deletion of Rankl in DMP1$^+$ lineage cells, referred to as

Rankl$^{-/-}$ herein) mice. Heterozygous Trap-Cre mice were crossed with Netrin-1$^{flox/flox}$; the offspring were intercrossed to generate the following genotypes: WT, Trap-Cre (mice expressing Cre recombinase driven by Trap promoter), Netrin-1$^{flox/flox}$ (mice homozygous for the Netrin-1 flox allele, referred to as Netrin-1$^{f/f}$ herein), Trap-Cre; Netrin-1$^{flox/flox}$ (conditional deletion of Netrin-1 in TRAP$^+$ lineage cells, referred to as Netrin-1$^{-/-}$ herein) mice. The genotypes of the mice were determined by PCR analyses of genomic DNA, which was extracted from mouse tails within the following primers: Avil-Cre: forward: 5′-CCCTGTTCACTGTGAG TAGG-3′, reverse: 5′-GCGATCCCTGAACATGTCCATC-3′, WT: 5′-AGTATCT GGTAGGTGCTTCCAG-3′; EP4 loxP allele forward: 5′-TCTGTGAAGCGAGTC CTTAGGCT-3′, reverse: 5′-CGCACTCTCTCTCTCCCAAGGAA-3′; Dmp1-Cre, forward: 5′-CCCGCAGAACCTGAAGATG-3′, reverse: 5′-GACCCGGCAAAAC AGGTAG-3′; Rankl loxP allele: forward: 5′-CTGGGAGCGCAGGTTAAATA-3′, reverse: 5′- GCCAATAATTAAAATACTGCAGGAAA-3′; Trap-Cre: forward: 5′-ATATCTCACGTACTGACGGTGGG-3′, reverse: 5′-CTGTTTCACTATCCAGG TTACGG-3′; Netrin-1 loxP allele: forward: 5′-AGGTAAAGTCTCCTACGCGG-3′, reverse: 5′-CTTCCAAACCTGAACCGCCC-3′. We performed LSI or sham surgery in 2-month-old male EP4$^{f/f}$, EP4$^{-/-}$, Rankl$^{f/f}$, Rankl$^{-/-}$, Netrin-1$^{f/f}$, and Netrin-1$^{-/-}$ mice. These mice were euthanized with an overdose of isoflurane inhalation at 4 or 8 weeks after LSI or sham surgery (12 per group). All mice were maintained at the animal facility of The Johns Hopkins University School of Medicine. All experimental protocols were approved by the Animal Care and Use Committee of The Johns Hopkins University, Baltimore, MD.

### μCT
Mice were euthanized with an overdose of isoflurane inhalation and flushed with phosphate-buffered saline (PBS) for 5 min followed by 10% buffered formalin perfusion for 5 min via the left ventricle. Then, the lower thoracic and whole lumbar spine were dissected and fixed in 10% buffered formalin for 48 h, transferred into PBS, and examined by high-resolution μCT (Skyscan1172). The scanner was set at a voltage of 55 kVp, a current of 181 μA, and a resolution of 9.0 μm per pixel to measure the endplates and vertebrae. The ribs on the lower thoracic spine were included for identification of L4–L5 unit localization. Images were reconstructed and analyzed using NRecon v1.6 and CTAn v1.9 (Skyscan US, San Jose, CA), respectively. Coronal images of the L4–L5 unit were used to perform three-dimensional histomorphometric analyses of the caudal endplate. Coronal images of the L5 vertebrae were used to perform three-dimensional histomorphometric analyses of the trabecular bone or cortical bone (anterior shell). Three-dimensional structural parameters analyzed were total porosity and Tb.Sp for the endplates, trabecular BV/TV, Tb.N, Tb.Th, and Tb. Sp for L5 vertebrae. Six consecutive coronal-oriented images were used for showing 3-dimensional reconstruction of the endplates and the vertebrae using three-dimensional model visualization software, CTVol v2.0 (Skyscan US).

### Histochemistry, immunohistochemistry, and histomorphometry
At the time of euthanasia, the L3–L5 lumbar spine and DRGs were collected and fixed in 10% buffered formalin for 48 h or overnight. Then, the spine samples were decalcified in 10% or 0.5 M EDTA (pH 7.4) for 14 or 5 days and embedded in paraffin or optimal cutting temperature (OCT) compound (Sakura Finetek, Torrance, CA). Four-μm-thick coronal-oriented sections of the L4–L5 lumbar spine were processed for Safranin O and fast green, TRAP (Sigma-Aldrich), and immunohistochemistry staining using a standard protocol. Thirty-μm-thick coronal-oriented sections were prepared for sensory nerve- and blood vessel-related immunofluorescent staining, and ten-μm-thick coronal-oriented sections were used for other immunofluorescent staining using a standard protocol. The sections were incubated with primary antibodies to mouse endomucin (1:50, sc-65495, Santa Cruz Biotechnology), CD31 (1:50, ab28364, Abcam), CGRP (1:100, ab81887, Abcam), PGP9.5 (1:100, ab10404, Abcam), DCC (1:100, ab201260, Abcam), Netrin-1 (1:100, ab39370, Abcam), TRAP (1:200, ab185716, Abcam), Cox-2 (1:100, ab15191, Abcam), EP4 (1:10, ab92763, Abcam), IB4 (1:100, I21411, Thermo Fisher Scientific, Waltham, MA), Na$_v$1.8 (1:100, ab93616, Abcam) PGE2 (1:100, ab2318, Abcam) overnight at 4 °C. Then, the corresponding secondary antibodies were added onto

the sections for 1 h while avoiding light. For immunohistochemistry, a horseradish peroxidase–streptavidin detection system (Dako) was subsequently used to detect the immunoactivity, followed by counterstaining with hematoxylin (Sigma-Aldrich). For immunofluorescent staining, the sections were counterstained with 4′,6-diamidino-2-phenylindole (DAPI, Vector, H-1200). The sample images were observed and captured by a fluorescence microscope (Olympus BX51, DP71) or confocal microscope (Zeiss LSM 780). ImageJ (NIH) software was used for quantitative analysis. We calculated endplate scores as described previously[66,67].

**Retrograde and anterograde tracing**. Two-month-old male C57BL/6J mice (Charles River) were used to perform LSI or sham surgery (6 per group). We anesthetized the mice with ketamine and xylazine at 8 weeks after surgery. For the retrograde nerve tracing experiment, the caudal endplate of L4–L5 was adequately exposed with a ventral approach. Then, 2 μL Dil (Molecular Probes; 2 mg/ml in methanol) was injected into the left part of caudal endplate of L4–L5 using borosilicate glass capillaries after drilling a hole at left part of endplate. The drilling holes were sealed with bone wax immediately after injection to prevent tracer leakage. After Dil injection, the wound was sutured, and a heating pad was used to protect mice during recovery from anesthesia. Mice were euthanized with an overdose of isoflurane inhalation, and the DRGs (T12–L6) were isolated for immunofluorescence at 1 week after Dil injection. Ten-μm-thick frozen sections were used, and the Dil signals were inspected under 564-nm excitation using a confocal microscope (LSM 780, Zeiss).

For the anterograde-tracing experiment, 2-month-old male C57BL/6J mice (Charles River) were used to perform LSI or sham surgery and 3- and 20-month-old male C57BL/6J mice (Jackson Laboratory) were prepared for the aging-induced endplate degeneration model (six per group). We anesthetized the aging model mice and the operated mice at 8 weeks after surgery with ketamine and xylazine. The L1 and L2 DRGs were adequately exposed with a dorsal approach. Then, 2 μL Dil (Molecular Probes; 2 mg/ml in methanol) was injected into the DRGs using borosilicate glass capillaries. After Dil injection, the wound was sutured, and a heating pad was used to protect mice during recovery from anesthesia. Mice were euthanized with an overdose of isoflurane inhalation, and the L3–L5 spine was collected for immunofluorescence at 1 week after Dil injection. Thirty-μm-thick coronal-oriented frozen sections were used, and the Dil signals were inspected under 564-nm excitation using a confocal microscope (LSM 780, Zeiss).

**qRT-PCR**. The total RNA was extracted from lumbar spinal endplate tissue samples using TRIzol reagent (Invitrogen, Carlsbad, CA) according to the manufacturer's instructions. The purity of RNA was tested by measuring the ratio of absorbance at 260 nm over 280 nm. For qRT-PCR, 1 μg RNA was reverse transcribed into complementary DNA using the SuperScript First-Strand Synthesis System (Invitrogen), then qRT-PCR was performed with SYBR Green-Master Mix (Qiagen, Hilden, Germany) on a C1000 Thermal Cycler (Bio-Rad Laboratories, Hercules, CA). Relative expression was calculated for each gene by the $2^{-\triangle\triangle CT}$ method, with glyceraldehyde 3-phosphate dehydrogenase for normalization. Primers used for qRT-PCR are listed in Supplementary Table 2.

**ELISA**. The concentrations of PGE2 and netrin-1 in the L3–L5 endplates were determined by using the PGE2 Parameter Assay Kit (KGE004B, R&D Systems) and Mouse Netrin-1 ELISA Kit (EKC37454, Biomatik, Wilmington, DE) according to the manufacturer's instructions (three per group), respectively.

**DRG neuron culture**. We processed DRG neuron culture as described previously[68]. Briefly, the dishes or coverslips for DRG neuron culture were coated with 500 μl working solution containing 100 μg/ml Poly-D-Lysine and 10 μg/ml Laminin at 37 °C. To prepare the neuron culture medium, we supplemented alpha minimum essential medium (α-MEM) with 1 × penicillin–streptomycin solution (500 units of penicillin and 500 μg of streptomycin, Gibco Laboratories, Gaithersburg, MD), 5% fetal bovine serum (Gibco), 1 × GlutaMAX-I supplement (35050-061, Thermo Fisher Scientific), and the antimitotic reagents containing 20 μM 5-fluoro-2-deoxyuridine (F0503, Sigma-Aldrich) and 20 μM uridine (U3003, Sigma-Aldrich). For the serum-free medium, the fetal bovine serum was replaced with the supplement B27. After euthanizing the 6- to 8-week-old mice, the lumbar DRGs were harvested and stored in microfuge tubes with α-MEM medium placed on ice. DRG neurons were digested and dissociated with 1 mg/ml collagenase A solution (10103578001, Roche, Basel, Switzerland) in a 37 °C incubator for 90 min followed by 1 × TrypLE Express solution (15140-122, Thermo Fisher) at 37 °C for 20 min. Then, TrypLE Express solution was removed, and DRGs were washed gently three times with 1 ml of prepared culture medium (containing 5% fetal bovine serum). Tissue was triturated by gently pipetting up and down 20–30 times in 1 ml of prepared culture medium. After trituration, the DRG neuron suspension was filtered (40-μm strainer) following non-dissociated tissue settlement to the bottom of the microfuge and transferred to another tube. After centrifugation (800 r.p.m for 4 min at room temperature), the cell pellet was resuspended and cultured in a precoated dish.

**Western blot**. The primary DRG neurons were treated with 20 μM PGE2 (14010, Cayman Chemical, Ann Arbor, MI) for 30 min; 10 μM PKA inhibitor (H-89, BML-E1196, Enzo Life Sciences, Farmingdale, NY) for 1 h. Western blot analysis was conducted on the protein lysates from the cultured primary DRG neurons. The supernatants of lysates were collected after centrifugation and separated by SDS-PAGE (sodium dodecyl sulfate-polyacrylamide gel electrophoresis) and then blotted on the nitrocellulose blotting membranes (Bio-Rad Laboratories). Specific antibodies were applied for incubation, and the proteins were detected by using an enhanced chemiluminescence kit (Amersham Bioscience, RNP2108). The antibodies used for western blot were p-CREB (1:1000, 9198, Cell Signaling Technology), CREB (1:1000, 9197, Cell Signaling Technology), p-PKA (1:1000, 5661, Cell Signaling Technology), PKA (1:1000, 4782 S, Cell Signaling Technology), and β-tubulin (1:2000, MA5-16308, Invitrogen). The original blots are provided in the Source Data file.

**Sodium indication**. For sodium indication, ANG-2 AM (Teflabs, Austin, TX) was used according to the manufacturer's protocol. Briefly, 1 mM stock solution of ANG-2 AM was diluted to twice the original volume with a solution of 20% Pluronic F-127 (Thermo Fisher Scientific) in DMSO. Then, the ANG-2 AM/Pluronic F-127 solution was dispersed to final concentration at 5 μM ANG-2 AM and 0.1% Pluronic F-127 with serum-free culture medium. After incubation for 1 h at 37 °C, the cell loading medium was removed. The cells were washed with serum-free and dye-free medium and prepared for sodium imaging. The sterile imaging buffer contained 5.4 mM KCl, 160 mM NaCl, 20 mM HEPES, 1.3 mM CaCl$_2$, 0.8 mM MgSO$_4$, 0.78 mM NaH$_2$PO$_4$, and 5 mM glucose (pH 7.4). The sodium imaging was monitored and captured using a confocal microscope (Zeiss LSM 780).

In different sets of experiments, the DRG neurons were treated with 20 μM PGE2 (14010, Cayman Chemical, Ann Arbor, MI) for 5 min, 10 μM PKA inhibitor (H-89, BML-E1196, Enzo Life Sciences, Farmingdale, NY) for 1 h, dibutyryl-cAMP (28945, Sigma-Aldrich) for 5 min, or siRNA oligo for Nav 1.8 (GE Healthcare Dharmacon, Lafayette, CO). The siRNA transfection was by using Lipofectamine RNAiMAX Transfection Reagent (Thermo Fisher Scientific) using a standard protocol.

For immunostaining, the DRG neurons were washed three times with PBS, followed by fixation by 4% paraformaldehyde (PFA) for 20 min at room temperature. The immunofluorescent staining used a standard protocol. The coverslips were incubated with primary antibodies to mouse CREB (1:100, 9197, Cell Signaling Technology), p-CREB (1:100, ab32096, Abcam), CGRP (1:100, ab81887, Abcam), PKA (1:100, 4782 Cell Signaling Technology), and p-PKA (1:100, ab227848, Abcam) overnight at 4 °C. Then, the corresponding secondary antibodies were added onto the coverslips for 1 h while avoiding light. The coverslips were counterstained with 4′,6-diamidino-2-phenylindole (DAPI, Vector, H-1200). The sample images were observed and captured using a microscope (Olympus BX51, DP71).

**Behavioral testing**. Behavioral testing was performed before surgery and weekly after surgery. All behavioral tests were performed by the same investigator, who was blinded to the study groups.

Vocalization thresholds in response to the force of an applied force gauge (SMALGO algometer; Bioseb, Pinellas Park, FL) were measured as pressure hyperalgesia[59]. A 5-mm-diameter sensor tip was directly pressed on the dorsal skin over L4–L5 (0.5 cm above the line connecting posterior iliac crest), while the mice were gently restrained. The pressure force was increased at 50 g/s until an audible vocalization was made. The curve of pressure force was recorded by using BIO-CIS software (Bioseb) to ensure the force increased gradually. A cutoff force of 500 g was used to prevent tissue trauma. Two tests were performed 15 min apart, and the mean value was calculated as the nociceptive threshold.

Spontaneous wheel-running activity was recorded using activity wheels designed for mice (model BIO-ACTIVW-M, Bioseb)[69]. The software enabled recording of activity in a cage similar to the mice's home cage, with dimensions of 35 × 20 × 13 cm, and the wheel (diameter: 23 cm, lane width: 5 cm) could be spun in both directions. The device was connected to an analyzer that automatically records the spontaneous activity. The mice had ad libitum access to food and water. We evaluated the distance traveled, mean speed, maximum speed, and total active time during 2 days for each mouse.

The hind paw withdrawal frequency in response to a mechanical stimulus was determined using von Frey filaments of 0.7 mN and 3.9 mN (Stoelting, Wood Dale, IL). Mice were placed on a wire metal mesh grid covered with a clear plastic cage. Mice were allowed to acclimatize to the environment for 30 min before testing. Von Frey filaments were applied to the mid-plantar surface of the hind paw through the mesh floor with enough pressure to buckle the filaments. Probing was performed only when the mouse's paw was in contact with the floor. A trial consisted of application of a von Frey filament to the hind paw 10 times at 1-s intervals. If withdraw occurred after application, it was recorded, and the next application was performed similarly when the mouse's paw was again in contact with the floor. Mechanical withdrawal frequency was calculated as the percentage of withdrawal times in response to ten applications.

Straight leg-raising test was performed by stretching the hindlimb (knee joint fully extended) and flexing the hip for 2 s. We recorded the number of vocalizations in five leg stretch-and-lifts[61,70]. The negative result indicates that the nerve root compression is not involved in the hyperalgesia developed after LSI surgery.

**Statistics**. All data analyses were performed using SPSS, version 15.0, software (IBM Corp.). Data are presented as means ± standard deviations. For comparisons between two groups, we used unpaired, two-tailed Student's $t$ tests. For comparisons among multiple groups, we used one-way ANOVA with Bonferroni's post hoc test. For all experiments, $p < 0.05$ was considered to be significant. All inclusion/exclusion criteria were preestablished, and no samples or animals were excluded from the analysis. No statistical method was used to predetermine the sample size. The experiments were randomized, and the investigators were blinded to allocation during experiments and outcome assessment. The same sample was not measured repeatedly.

**Reporting summary**. Further information on research design is available in the Nature Research Reporting Summary linked to this article.

## Data availability

All relevant data that support the findings of this study are available within this published article (and its Supplementary Information files or available from the corresponding author upon reasonable request. The source data underlying Figs. 1a–m, 2b, 3b, c, e, f, h, 4c, e, 4g, i, 5a, c, f, i, k, l, 6b–e, h–k, 7a–g, 8b, c, e, g, i–p, 9b, e, g, and 9i–p and Supplementary Figs. 1b, 2b–e, 3b, 4b, 6b, 7b–e, 8b, 9b, 10b, and 10b are provided as a Source Data file.

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

## Acknowledgements

This research was supported by Fox Gift (to X.C.). The authors thank Rachel Box, Jenni Weems, and Kerry Kennedy in the editorial office at the Department of Orthopaedic Surgery, The Johns Hopkins University, for editing the paper.

## Author contributions

S.N. conceived the experimental designs, conducted most of the experiments, and prepared the paper. Z.L. helped with behavior analysis, animal surgery, and histology sections. X.W. provided some ideas for the experiments and helped write the paper. Y.C. and T.W. provided some of the human samples and helped with human sample histology. R.D. provided suggestions and helped write the paper. J.C., R.S., S.D., G.Z., and A.J. provided suggestions for the project. P.W., D.P., B.H., X.L., Y.L., H.C., and H.Q. helped with animal surgery and behavioral tests. Y.G. and X.D. helped with behavior analysis. M.W., X.Z., and H.L. provided suggestions for the project. J.H. provided suggestions and proofread the paper. X.C. conceived the idea, supervised the project, and wrote most of the paper.

## Competing interests

The authors declare no competing interests.
