## [Peer Review File · Nature Communications]

Reviewers' comments:

Reviewer #1 (Remarks to the Author):

The authors have submitted a body of work which I believe will positively impact the field.

Title: that osteoclasts are inducing sensory innervation in vertebral endplates is highly likely but could the authors be more categorical in their title wrt a mechanism of action?

Abstract

'Spinal pain is one of the most common health problems'. As a standalone comment this sounds subjective.

'Aging induces behaviour associated with spinal pain' – what behaviour? Please clarify.

The sentence (line)32 should be revised as the second half of the sentence is unclear.

Sentence beginning line 35 – did the authors show this? If so, state 'we showed that..'

Introduction

The first 4 sentences should be condensed.

Line 55 The authors can only state that disc degeneration is not painful in some patients – they do not know if it is true for all.

Line 58 – Tell us straight away what the relationship is between the Modic changes and LBP.

Please re-read the introduction in full as grammatical errors are apparent throughout.

Behavioural testing for mechanical allodynia – why were only 2 forces used?

Statistical analysis: No sample size is reported in the abstract. Can the authors confirm potential bias due to lost, missing or excluded data? Can the authors include a statement about outliers?

Ensure that the ARRIVE checklist has been adhered to. The experimental unit is not always clear.

Were randomisation methods used to assign animals to experimental conditions? Blinding? Each experimental group should have a clear section relating to HOW each experiment was carried out (i.e. drug dose or method of euthanasia).

What methods were used to allocate animals to experimental groups?

Reviewer #2 (Remarks to the Author):

Low back pain (LBP) is one of the most common chronic pain conditions that causes significant social and economic burden, and degenerative disc is considered as the major source of LBP. In this report, Ni et al demonstrated that porous endplate, rather than the disc, is a major source of LBP. Specifically, they showed increased osteoclast activity in endplates of lumbar spine instability (LSI) mouse model and in aged mice, which induces CGRP (+) neuron innervation of porous endplates. The porous endplates produce PGE2 to activate CGRP (+) nerve fibers by PGE2 receptor EP4, and eventually leads to sodium influx through Nav1.8 channel. Inhibition of EP4 from DRG sensory neurons, inhibition of osteoclast formation by knocking out Rank1, or preventing sensory innervation into porous endplates by knocking out Nestrin-1 from osteoclasts greatly attenuate pain behavior induced by LSI. This study provides a new mechanism of LBP, and potentially has significant impact in the field. I only have a few moderate concerns and some minor concerns about this paper.

Moderate concerns:

1. Although vocalization threshold in response to force applied can be considered as a pain behavior of LBP, the authors should clarify that all the testes of spontaneous activities are not specific pain behaviors. Lumbar spine instability itself, with damage of the spine, can potentially

reduce spontaneous activities without pain. Similarly, aged mice can have reduced spontaneous activities without pain.

2. It is hard to understand why LSI mice can have hind paw mechanical hypersensitivity, which is usually the result from hind paw local inflammation or nerve injury. In LSI model, the injury is far away from hind paw, and without local inflammation or nerve injury (with negative straight leg raising test), the authors should discuss how hind paw mechanical hypersensitivity can develop in LSI model.

3. In pathological study of human samples, the subjects with LBP are significantly older than subjects without LBP (Supplementary Table). The authors should clarify this key difference in main text. Moreover, because the age has significant impact on endplate pathology, as demonstrated in mouse study, the authors should tone down their claim that the high endplate scores is linked to history of frequent LBP.

Minor concerns:

1. Although porous endplates might be one of the major sources of LBP, it is unclear it is the "primary" source of LBP, as claimed in line 53.

2. "TRAP" in Fig 2A is labeled as red, not magenta.

3. It would be very helpful to include three-dimensional high-resolution uCT images and safranin O/fast green staining of endplates in Fig 2.

4. Fig 4F shows some nerve fibers in deeper layer. The authors should outline or label the endplate to make it clear.

5. It would be great if the authors can show which type of the cells make COX2 and PGE2 in the porous endplates

6. Fig 5G actually shows that many Nav1.8 signals does NOT overlap with CGRP. The authors need to present a better image to demonstrate the colocalization.

7. For Fig 5J-L, it would be much better to include histograms in addition to the images. The authors also need to clarify how long the cells were treated with PGE2.

8. Fig 5L needs EP4 f/f with PGE2 only as a positive control

9. It would be interesting to know if EP4 -/- has any effect on osteoclast

10. The function of angiogenesis is unclear in LBP and therefore seems to be irrelevant to the main goal of this study. It might be better to move these results to supplementary data.

Reviewer #3 (Remarks to the Author):

Low back pain (LBP) associated with Degenerative disc disease is a serious clinical problem with no effective treatment. The underlying mechanism of LBP is still largely unknown. This manuscript attempts to study the cellular and molecular mechanisms of LBP using two mouse models: aging and lumbar spine instability surgery. Analysis of sclerotic endplates in these models revealed that PGE2 activates sensory nerves to induce spinal pain behavior and that osteoclasts play a primary role in promoting sensory innervation through secreting Netrin-1. To my knowledge, this is the first

study proposing that remodeling of endplate via endochondral ossification is the major cause of LBP. Therefore, it represents an important breakthrough in delineating the mechanisms of LBP. However, the quality of data, especially the imaging data, somewhat diminish the enthusiasm for this manuscript.

Major comments:

1. Most imaging data are shown at a high magnification. It would be better to show at least some of them at a low magnification in order to orientate the readers and to display internal controls. For example, in Fig. 1A, the image should include some disc area and some vertebral bone area. The authors proposed that after LSI endplates become a bone like structure containing osteoblasts, osteoclasts, and bone marrow. Since osteoclast is the major cause for pain, it would be interesting to see whether sclerotic endplates have more osteoclasts than vertebral bone. The same applies to Fig. 5B. For uCT images, only sclerotic endplates were shown. How about the underlying vertebral trabecular bone? Is there any change there?
2. Some images, such as Fig. 2F and 4B, are totally black on paper. When zoomed in on the computer screen, it appears that Fig. 2F LSI panels do have some green signals of IB4 and that Fig. 4B LSI L3 do not have any red signal, both of which are not consistent with authors' conclusion.
3. Some images do not match with their quantification data. Fig. 4D L2-L top panel: I do see many Dil+ only cells but quantification in E says 100% Dil+ cells are CGRP+. The label for red square in E should be Dil+CGRP+ cells %. Fig. 5E and F also do not match. There were very few yellow cells in all 4 images.
4. Fig. 5J, L: images seem to be at a low resolution and not focused.
5. Fig. 5K: Western blot is a better way to demonstrate changes of p-PKA and p-CREB in cell culture than immunofluorescence.
6. Fig. 7A: does Rank1 CKO affect the bony structure in endplates without surgery? Again, showing the underlying vertebral bone will serve as a positive control that CKO does have osteopetrosis phenotype.
5. Two mechanisms are proposed to explain LBP: PGE2/EP4/nerve and osteoclast/Netrin-1/nerve. Are they interconnected? Are osteoclasts the major source of PGE2 in endplates? From Fig. 5B, it seems that most bone marrow cells are PGE2 and Cox2 positive.

Minor comments:

1. Type H vessels, as proposed by Adams group in 2014, refer to CD31^{high}Emcn^{High} vessels. All bone marrow capillaries express CD31 and Emcn. Therefore, the authors cannot call their CD31⁺Emcn⁺ cells as type H vessels.
2. Osteoclast number should be normalized against tissue area.

We wanted to thank the reviewers for their thoughtful and constructive comments for our manuscript. We have addressed all the comments with additional experimentation and clarification. Detailed point-to-point responses to reviewers' critiques are described as below, and the revisions are highlighted in the manuscript.

Reviewers' comments:

Reviewer #1 (Remarks to the Author):

The authors have submitted a body of work which I believe will positively impact the field.

Response: We are encouraged by the reviewer's overall insightful comments.

Title: that osteoclasts are inducing sensory innervation in vertebral endplates is highly likely but could the authors be more categorical in their title wrt a mechanism of action?

Response: We appreciate the reviewer's suggestion. The title has been changed as 'Sensory Innervation in Porous Endplates by Netrin-1 from Osteoclasts Mediates PGE2-Induced Spinal Pain' in the revised manuscript.

Abstract

'Spinal pain is one of the most common health problems'. As a standalone comment this sounds subjective.

'Aging induces behaviour associated with spinal pain' – what behaviour? Please clarify.

The sentence (line)32 should be revised as the second half of the sentence is unclear.

Sentence beginning line 35 – did the authors show this? If so, state 'we showed that..'

Response: We appreciate the reviewer's valuable comment. We have changed it to "Spinal pain is the leading cause of activity limitation and work absence as a common health problem, however, its origins and underlying mechanisms remain unclear." in the revised manuscript. The spinal pain-associated behaviors include pressure hyperalgesia, spontaneous activity, mechanical hyperalgesia of the hind paw in the present study. Because the abstract can not exceed 150 words, we could not include detailed information in the Abstract. The detailed description of the spinal pain-associated behaviors has been included in the Method. For the sentence (line 32), it has been rewritten as "Here we report that osteoclasts induce sensory innervation in the porous endplates for the spinal pain". For the sentence beginning line 35, 'we show that' has been added in the revised manuscript.

Introduction

The first 4 sentences should be condensed.

Response: We appreciate the reviewer's suggestion. The first 4 sentences have been condensed in the revised manuscript.

Line 55 The authors can only state that disc degeneration is not painful in some patients – they do not know if it is true for all.

Response: We appreciate the reviewer's precise suggestion. 'in some patients' has been added at the end of this sentence in the revised manuscript.

Line 58 – Tell us straight away what the relationship is between the Modic changes and LBP.

Response: We appreciate the reviewer's suggestion. We have revised this sentence as 'the positive association between....' in the revised manuscript.

Please re-read the introduction in full as grammatical errors are apparent throughout.

Response: We appreciate the reviewer's important suggestion and apologize for the errors. The errors have been corrected in the revised manuscript.

Behavioral testing for mechanical allodynia – why were only 2 forces used?

Response: We appreciate the reviewer's comment. We used withdrawal frequency of hind paw to show mechanical allodynia. The stimuli were applied to the hind paws 10 times, and the withdrawal frequencies of hind paw were recorded. A low intensity- and a moderate intensity-filament were commonly selected for the testing. The two forces used in our manuscript are 0.7mN (low intensity) and 3.9mN (moderate intensity).

Statistical analysis: No sample size is reported in the abstract. Can the authors confirm potential bias due to lost, missing or excluded data? Can the authors include a statement about outliers? Ensure that the ARRIVE checklist has been adhered to. The experimental unit is not always clear. Were randomisation methods used to assign animals to experimental conditions? Blinding? Each experimental group should have a clear section relating to HOW each experiment was carried out (i.e. drug dose or method of euthanasia).

What methods were used to allocate animals to experimental groups?

Response: We appreciate the reviewer's valuable comments. We have reported the sample size for each experiment in the figure legends in the revised manuscript. We have stated that no samples or animals were excluded from the analysis; no statistical method was used to predetermine the sample size; the experiments were randomized, and the investigators were blinded to allocation during experiments and outcome assessment in the Statistics part. The ARRIVE checklist has been adhered. We anesthetized the mice with ketamine (Vetalar, Ketaset, Ketalar; 100 mg/kg, intraperitoneally) and xylazine (Rompun, Sedazine, AnaSed; 10 mg/kg, intraperitoneally) to conduct surgery. For retrograde and anterograde tracing experiments, 2 μ L Dil (Molecular Probes; 2 mg/ml in methanol) was injected into the endplate or DRG. The dose of PGE2 (14010, Cayman Chemical, Ann Arbor, MI) and PKA inhibitor (H-89, BML-E1196, Enzo Life Sciences, Farmingdale, NY) is 20 μ M and 10 μ M for *in vitro* experiment. Mice were euthanized with an overdose of isoflurane (Forane, Baxter) inhalation at the corresponding time points to collect the samples. We have included detailed information in

the revised manuscript.

Reviewer #2 (Remarks to the Author):

Low back pain (LBP) is one of the most common chronic pain conditions that causes significant social and economic burden, and the degenerative disc is considered as the major source of LBP. In this report, Ni et al demonstrated that porous endplate, rather than the disc, is a major source of LBP. Specifically, they showed increased osteoclast activity in endplates of lumbar spine instability (LSI) mouse model and in aged mice, which induces CGRP (+) neuron innervation of porous endplates. The porous endplates produce PGE2 to activate CGRP (+) nerve fibers by PGE2 receptor EP4, and eventually leads to sodium influx through Nav1.8 channel. Inhibition of EP4 from DRG sensory neurons, inhibition of osteoclast formation by knocking out Rank1, or preventing sensory innervation into porous endplates by knocking out Netrin-1 from osteoclasts greatly attenuate pain behavior induced by LSI. This study provides a new mechanism of LBP, and potentially has significant impact in the field. I only have a few moderate concerns and some minor concerns about this paper.

Response: We are encouraged by the reviewer's insightful and accurate comments.

Moderate concerns:

1. Although vocalization threshold in response to force applied can be considered as a pain behavior of LBP, the authors should clarify that all the testes of spontaneous activities are not specific pain behaviors. Lumbar spine instability itself, with damage of the spine, can potentially reduce spontaneous activities without pain. Similarly, aged mice can have reduced spontaneous activities without pain.

Response: We appreciate the reviewer's valuable suggestions. The monitoring of spontaneous activities were used to evaluate several pain models, including inflammatory pain (Pain. 2012 Apr;153(4):876-84. doi: 10.1016/j.pain.2012.01.016.), neuropathic pain (Pain Res Manag. 2018 Jun 3;2018:8217613. doi: 10.1155/2018/8217613.), and osteoarthritis pain (Pharmacol Biochem Behav. 2011 Mar;98(1):35-42. doi: 10.1016/j.pbb.2010.12.009. Ref 61). We conducted tests of spontaneous activity to indicate the potential effect of spinal pain. We agree with the reviewer's comment about these tests are not specific for spinal pain. We have clarified that we monitored spontaneous activity to indicate the potential effect of spinal pain, but all the tests of spontaneous activity are not specific pain behaviors in the Results part.

2. It is hard to understand why LSI mice can have hind paw mechanical hypersensitivity, which is usually the result from hind paw local inflammation or nerve injury. In LSI model, the injury is far away from hind paw, and without local inflammation or nerve injury (with negative straight leg raising test), the authors should discuss how hind paw mechanical hypersensitivity can develop in LSI model.

Response: We appreciate the reviewer's important suggestions. The sign of the hind paw mechanical allodynia is considered as the secondary indicator of spinal pain-associated behaviors. Several studies reported that the hind paw mechanical allodynia develops in low back

pain animal models as the secondary hypersensitivity (ref 58-61)). Among these works of literature, one study about the lumbar facet joint osteoarthritis-induced spinal pain excluded the local inflammation or nerve injury (with negative straight leg raising test) (ref 61).

One study demonstrated that the mouse sciatic nerve predominantly originates from the L3 and L4 DRG by injecting retrograde labeling in the hind paw (ref 62). Our retrograde tracing data demonstrated that L3 DRG is also the partial origin of sensory nerves in the endplates of L4/5 in LSI mice (Fig. 4B, 4C). In addition, the dorsal horn of spinal cord receives inputs from several segmental DRGs (ref 63, 64). The major monosynaptic input for the dorsal horn neurons in L4 segment is from the L4–L6 DRGs, the dorsal horn neurons in L3 segment is from the L2–L5 DRGs (ref 65). These anatomical features might be the basis of the hind paw mechanical hypersensitivity in the LSI model, indicating the development of the referred pain. Referred pain is pain perceived at a location other than the site of the painful origin. It is caused by the nociceptive dorsal horn and brain stem neurons that receive convergent inputs from various tissues. Clinically, low back pain can be accompanied by referred pain in the lower extremities in patients without sciatica. We have included it in the Discussion part.

3. In pathological study of human samples, the subjects with LBP are significantly older than subjects without LBP (Supplementary Table). The authors should clarify this key difference in the main text. Moreover, because the age has significant impact on endplate pathology, as demonstrated in mouse study, the authors should tone down that the high endplate scores is linked to history of frequent LBP.

Response: We appreciate the reviewer's important suggestion. We have clarified the difference of age between patients with LBP and patients without LBP in the main text. Also, we toned down the statement that high endplate scores are linked to a history of frequent LBP. This sentence was rewritten as 'The increased endplate scores were also observed in patients with a history of frequent LBP (Supplementary Figure 4B). However, the patients with the history of frequent LBP are older than the ones without the history of frequent LBP (Supplementary Table 1)' in the revised manuscript.

Minor concerns:

1. Although porous endplates might be one of the major sources of LBP, it is unclear it is the "primary" source of LBP, as claimed in line 53.

Response: We appreciate the reviewer's valuable suggestion. We deleted this word in the revised manuscript.

2. "TRAP" in Fig 2A is labeled as red, not magenta.

Response: We appreciate the reviewer's comment. We have changed the color of "TRAP" in fig. 2A in the revised manuscript.

3. It would be very helpful to include three-dimensional high-resolution uCT images and safranin O/fast green staining of endplates in Fig 2.

Response: We appreciate the reviewer's comment. The sclerosis of endplates induced by the lumbar spine instability in wild type mice has been demonstrated by μ CT analysis and safranin O/fast green staining in our previous paper (ref 20). This paper demonstrated that the excessive activation of TGF beta signaling contributes to the LSI-induced endplate sclerosis.

4. Fig 4F shows some nerve fibers in deeper layer. The authors should outline or label the endplate to make it clear.

Response: We appreciate the reviewer's constructive suggestion. We have outlined the endplate with dash lines in Fig. 4F and Fig. 4H in the revised manuscript.

5. It would be great if the authors can show which type of the cells make COX2 and PGE2 in the porous endplates

Response: We appreciate the reviewer's important suggestion. Based on our immunostaining results of COX2 and PGE2 in Fig. 5B, many bone marrow cells in porous endplates are COX2 and PGE2 positive. Based on literature, osteoclast and osteoblast are the potential sources of PGE2 (Blood. 2005 Aug 15;106(4):1240-5. Nat Commun. 2019 Jan 14;10(1):181.). To show the source of PGE2 accumulated in the porous endplates, we conducted the co-immunostaining for COX2 with F4/80, COX2 with osteocalcin (OCN), and COX2 with TRAP. The results demonstrated that the COX2 is co-localized with F4/80⁺, some OCN⁺, and a few TRAP⁺ cells (Supplementary Figure. 5). These data showed that several types of cells in the porous endplates might be the source of PGE2 which accumulates in the porous endplates and stimulates the sensory nerves.

6. Fig 5G actually shows that many Nav1.8 signals does NOT overlap with CGRP. The authors need to present a better image to demonstrate the colocalization.

Response: We appreciate the reviewer's critical comment and apologized for the inappropriate image. We have presented a better image in the revised manuscript.

7. For Fig 5J-L, it would be much better to include histograms in addition to the images. The authors also need to clarify how long the cells were treated with PGE2.

Response: We appreciate the reviewer's valuable suggestion. We have included histograms in the new Fig 6B-C and Fig 6H-K in the revised manuscript. The cells were treated with 20 μ M PGE2 for 5 mins. We have included the detailed information in the Method and figure legend of the revised manuscript.

8. Fig 5L needs EP4 f/f with PGE2 only as a positive control

Response: We appreciate the reviewer's valuable suggestion. We have added the EP4^{f/f} with PGE2 treatment group as the control in the new Fig 6G in the revised manuscript.

9. It would be interesting to know if EP4 ^{-/-} has any effect on osteoclast

Response: We appreciate the reviewer's constructive comment. we conducted TRAP staining on the sections of the EP4 ^{f/f} mice and EP4 ^{-/-} mice at 8 weeks post LSI or sham surgery. TRAP staining results demonstrated that the EP4 ^{-/-} did not significantly affect the number of TRAP ⁺ osteoclasts in porous endplates either in the sham group or LSI group (Supplementary Figure. 6).

10. The function of angiogenesis is unclear in LBP and therefore seems to be irrelevant to the main goal of this study. It might be better to move these results to supplementary data.

Response: We appreciate the reviewer's important comment. The blood vessels usually are accompanied by nerves. The angiogenesis in the porous endplates indirectly indicated the nerve innervation. We have moved the results about angiogenesis to supplementary figure. 1, 3, 8, and 10 in the revised manuscript.

Reviewer #3 (Remarks to the Author):

Low back pain (LBP) associated with Degenerative disc disease is a serious clinical problem with no effective treatment. The underlying mechanism of LBP is still largely unknown. This manuscript attempts to study the cellular and molecular mechanisms of LBP using two mouse models: aging and lumbar spine instability surgery. Analysis of sclerotic endplates in these models revealed that PGE2 activates sensory nerves to induce spinal pain behavior and that osteoclasts play a primary role in promoting sensory innervation through secreting Netrin-1. To my knowledge, this is the first study proposing that remodeling of endplate via endochondral ossification is the major cause of LBP. Therefore, it represents an important breakthrough in delineating the mechanisms of LBP. However, the quality of data, especially the imaging data, somewhat diminish the enthusiasm for this manuscript.

Response: We appreciate the reviewer's overall insightful, constructive, and accurate comments. We have improved the quality of data in the revised manuscript.

Major comments:

1. Most imaging data are shown at a high magnification. It would be better to show at least some of them at a low magnification in order to orientate the readers and to display internal controls. For example, in Fig. 1A, the image should include some disc area and some vertebral bone area. The authors proposed that after LSI endplates become a bone like structure containing osteoblasts, osteoclasts, and bone marrow. Since osteoclast is the major cause for pain, it would be interesting to see whether sclerotic endplates have more osteoclasts than vertebral bone. The same applies to Fig. 5B. For μ CT images, only sclerotic endplates were shown. How about the underlying vertebral trabecular bone? Is there any change there?

Response: We appreciate the reviewer's important suggestion. We have replaced the images in Fig. 2A and Fig. 5B with low magnification images to show some disc area and some vertebral bone area as internal controls in the revised manuscript. For the μ CT data in Fig. 3 and Fig. 8, we included additional analysis of the vertebral trabecular bone in Supplementary Figure. 2 and Supplementary Figure. 7 in the revised manuscript. The μ CT analysis demonstrated that the trabecular bone volume/total volume (BV/TV) and trabecular bone number (Tb.N) of L5 vertebrae decreased significantly in 20-month-old mice relative to 3-month-old mice (Supplementary Figure. 2). Also, the trabecular BV/TV, Tb.N, trabecular bone thickness (Tb.Th) increased and trabecular bone separation distribution (Tb. Sp) decreased significantly in RANKL^{-/-} mice relative to RANKL^{f/f} mice. However, there is no significant difference in these parameters between sham surgery group and LSI surgery group in RANKL^{-/-} mice or RANKL^{f/f} mice (supplementary fig. 7).

2. Some images, such as Fig. 2F and 4B, are totally black on paper. When zoomed in on the computer screen, it appears that Fig. 2F LSI panels do have some green signals of IB4 and that Fig. 4B LSI L3 do not have any red signal, both of which are not consistent with authors'

conclusion.

Response: We appreciate the reviewer's important suggestion and apologize for the inappropriate representative images. Because Fig. 2F were overexposed, the green signals were background signals. For Fig. 4B LSI L3, there were red signals of Dil in the original image. However, the image quality in the merged PDF is poor leading to the loss of information. We have replaced these images with more clear and representative ones in the revised manuscript.

3. Some images do not match with their quantification data. Fig. 4D L2-L top panel: I do see many Dil⁺ only cells but quantification in E says 100% Dil⁺ cells are CGRP⁺. The label for red square in E should be Dil⁺CGRP⁺ cells %. Fig. 5E and F also do not match. There were very few yellow cells in all 4 images.

Response: We appreciate the reviewer's important suggestion and apologize for the inappropriate representative images. Dil⁺ signals are co-localized with CGRP⁺ in the previous Fig. 4D L2-L top panel. But the image quality got poor in the merged PDF, and the positive signals became unclear. We have replaced the inappropriate images in Fig. 4D and Fig. 5E with more precise and representative images in the revised manuscript. For the quantification in Fig. 4E, we showed the percentage of Dil⁺CGRP⁺ cells in the total Dil⁺ cells.

4. Fig. 5J, L: images seem to be at a low resolution and not focused.

Response: We appreciate the reviewer's important suggestion. The images in Fig. 5J, L were selected from time-lapse photos, which might reduce the image resolution. Besides, the image quality got poor in the merged PDF. We have replaced and optimized the images in Fig. 5J, L in the revised manuscript.

5. Fig. 5K: Western blot is a better way to demonstrate changes of p-PKA and p-CREB in cell culture than immunofluorescence.

Response: We appreciate the reviewer's valuable suggestion. To validate the changes of pPKA and pCREB, we did western blot to examine the level of pPKA and pCREB in the primary sensory neurons after treatment in the revised manuscript. Western blot demonstrated that the PKA inhibitor or EP4^{-/-} significantly reduced PGE2-induced increase in the level of pPKA and PCREB (Fig. 6D and 6E in the revised manuscript). These western blot data further validated the immunofluorescence data.

6. Fig. 7A: does Rankl CKO affect the bony structure in endplates without surgery? Again, showing the underlying vertebral bone will serve as a positive control that CKO does have osteopetrosis phenotype.

Response: We appreciate the reviewer's valuable suggestion. From the μ CT analysis and safranin O/fast green staining, RANKL CKO did not significantly affect the structure of endplate without surgery. To demonstrate the osteopetrosis phenotype in RANKL CKO mice, the uCT analysis of the L5 vertebrae was conducted in the revised manuscript (Supplementary Figure. 7). The uCT

results demonstrated that the trabecular BV/TV, Tb.N, and Tb.Th increased and trabecular bone separation distribution (Tb.Sp) decreased significantly in the L5 vertebrae of RANKL CKO mice relative to RANKL^{f/f} mice.

7. Two mechanisms are proposed to explain LBP: PGE2/EP4/nerve and osteoclast/Netrin-1/nerve. Are they interconnected? Are osteoclasts the major source of PGE2 in endplates? From Fig. 5B, it seems that most bone marrow cells are PGE2 and Cox2 positive.

Response: We appreciate the reviewer's thoughtful comment. In our manuscript, the osteoclast-secreted Netrin-1 attracts the growth of sensory nerves into the porous endplates. The PGE2 accumulated in the porous endplates stimulates the newly innervated nerves through its receptor EP4 and causes spinal pain. The immunostaining results in Fig. 5B showed that many bone marrow cells in porous endplates are COX2 and PGE2 positive. Some studies stated that the osteoclast and osteoblast are the potential sources of PGE2 (Blood. 2005 Aug 15;106(4):1240-5. Nat Commun. 2019 Jan 14;10(1):181.). To show the potential source of PGE2 in the porous endplates, we conducted the co-immunostaining for COX2 with F4/80, COX2 with TRAP, COX2 with osteocalcin (OCN). The results of immunofluorescent staining demonstrated that the COX2⁺ cells are co-localized with F4/80⁺, some OCN⁺, and a few TRAP⁺ cells in the porous endplates (Supplementary Figure. 5). These data showed that several types of cells in the porous endplates might be the source of PGE2 which accumulates in the porous endplates and stimulates the sensory nerves.

Minor comments:

1. Type H vessels, as proposed by Adams group in 2014, refer to CD31^{high}Emcn^{High} vessels. All bone marrow capillaries express CD31 and Emcn. Therefore, the authors cannot call their CD31⁺Emcn⁺ cells as type H vessels.

Response: We appreciate the reviewer's important comment. We deleted the claim of type H vessels in the revised manuscript. We claimed the blood vessels in porous endplate as CD31⁺Emcn⁺ vessels.

2. Osteoclast number should be normalized against tissue area.

Response: We appreciate the reviewer's important suggestion. We have normalized osteoclast number against tissue area in the revised manuscript.

REVIEWERS' COMMENTS:

Reviewer #1 (Remarks to the Author):

I would like to thank the authors for their thoughtful and thorough corrections relating to my earlier queries. The additions have enabled a clearer understanding of the experimental questions and protocols.

Reviewer #2 (Remarks to the Author):

No more comments. All the concerns have been well addressed.

Reviewer #3 (Remarks to the Author):

The authors have adequately addressed my previous comments. I have no further questions.

REVIEWERS' COMMENTS:

Reviewer #1 (Remarks to the Author):

I would like to thank the authors for their thoughtful and thorough corrections relating to my earlier queries. The additions have enabled a clearer understanding of the experimental questions and protocols.

Response: We are encouraged by the reviewer's comments. We appreciate the reviewer's efforts to review our manuscript.

Reviewer #2 (Remarks to the Author):

No more comments. All the concerns have been well addressed.

Response: We appreciate the reviewer's efforts to review our manuscript.

Reviewer #3 (Remarks to the Author):

The authors have adequately addressed my previous comments. I have no further questions.

Response: We appreciate the reviewer's efforts to review our manuscript.